# Three-Dimensional Cell Cultures: The Bridge between In Vitro and In Vivo Models

**DOI:** 10.3390/ijms241512046

**Published:** 2023-07-27

**Authors:** Ornella Urzì, Roberta Gasparro, Elisa Costanzo, Angela De Luca, Gianluca Giavaresi, Simona Fontana, Riccardo Alessandro

**Affiliations:** 1Department of Biomedicine, Neuroscience and Advanced Diagnostics (Bi.N.D), Section of Biology and Genetics, University of Palermo, 90133 Palermo, Italy; ornella.urzi@unipa.it (O.U.); roberta.gasparro@unipa.it (R.G.); elisa.costanzo01@unipa.it (E.C.); riccardo.alessandro@unipa.it (R.A.); 2IRCCS Istituto Ortopedico Rizzoli, SC Scienze e Tecnologie Chirurgiche, 40136 Bologna, Italy; angela.deluca@ior.it (A.D.L.); gianluca.giavaresi@ior.it (G.G.)

**Keywords:** 3D cell culture, bone, brain, heart, liver, lung, skin, cancer, regenerative medicine

## Abstract

Although historically, the traditional bidimensional in vitro cell system has been widely used in research, providing much fundamental information regarding cellular functions and signaling pathways as well as nuclear activities, the simplicity of this system does not fully reflect the heterogeneity and complexity of the in vivo systems. From this arises the need to use animals for experimental research and in vivo testing. Nevertheless, animal use in experimentation presents various aspects of complexity, such as ethical issues, which led Russell and Burch in 1959 to formulate the 3R (Replacement, Reduction, and Refinement) principle, underlying the urgent need to introduce non-animal-based methods in research. Considering this, three-dimensional (3D) models emerged in the scientific community as a bridge between in vitro and in vivo models, allowing for the achievement of cell differentiation and complexity while avoiding the use of animals in experimental research. The purpose of this review is to provide a general overview of the most common methods to establish 3D cell culture and to discuss their promising applications. Three-dimensional cell cultures have been employed as models to study both organ physiology and diseases; moreover, they represent a valuable tool for studying many aspects of cancer. Finally, the possibility of using 3D models for drug screening and regenerative medicine paves the way for the development of new therapeutic opportunities for many diseases.

## 1. Introduction

Cell culture techniques allow cells isolated from a tissue to grow and be propagated in vitro under controlled conditions (temperature, pH, oxygen, growth factors) [1], thus reducing the statistical variance and making the experimental replicates very similar. Although most of the findings obtained by using these in vitro systems have provided fundamental information regarding cellular functions and signaling pathways as well as nuclear activities, cell cultures remain a highly limited model. Cells grown as a monolayer do not reflect the in vivo microenvironment since cell–cell contacts are limited, and the tissue-specific architecture is lacking. Moreover, the interaction of cells with a proper extracellular matrix (ECM), which is also a rich reservoir of growth factors and bioactive molecules, strongly alters several biological processes, such as cell proliferation or differentiation [2]. For these reasons, in the last decades, three-dimensional (3D) cell cultures were developed. The advantages of using 3D cell cultures are listed in Table 1. First, whereas cells in 2D grow in flat monolayers, a 3D culture allows cells to grow and interact with the surrounding extracellular environment in three dimensions, more accurately reflecting what normally happens within the tissues of the living organisms. In the 3D models, cell–cell and cell–ECM interactions are promoted, thereby ensuring cell proliferation and differentiation; in addition, cell morphology, behaviour, and topology more faithfully mirror the in vivo conditions [3]. Another important difference between 2D and 3D cell cultures is represented by the access to nutrients—while cells grown in 2D have homogeneous availability of nutrients, 3D models spontaneously create a gradient of nutrients since cells embedded in the mass have less access to them [4]. In addition, 3D cell cultures allow cell propagation without the need for immortalization [5,6]. This feature is essential to maintaining the integrity of critical suppressor genes, such as ARF, INK4A, and TP53 [7], which are of interest in research focused on tumor initiation. Finally, another advantage of the use of 3D cell cultures is the possibility of combining them with a mouse model, for instance, performing the gene editing ex vivo in patient-derived organoids which then are injected into mice. This procedure will definitively reduce the amount of time needed to generate new transgenic mouse strains carrying tissue-specific mutations [8]. On the other hand, 3D cell cultures carry some disadvantages, for instance, the difficulties related to finding the right assay for downstream analyses [6]. Moreover, some currently used protocols for 2D cell cultures have to be revisited for 3D models, such as immunofluorescence, in which problems related to the penetration of the staining as well as the clearance of the samples have been described [9].

One of the first 3D models that has been developed was represented by spheroids. In 1970, Sutherland and colleagues set up a multicellular spheroid culture to achieve a functional phenotype of cancer cells and to study their response to radiotherapy [10]. Spheroids are simple cellular aggregates generated by either a single cell type or a multicellular mixture of cells, which are typically cultured as free-floating aggregates [11]. Spheroids contain a layer of cells exposed to the external environment, while other layers are embedded inside the sphere, creating a gradient in nutrients and oxygen that simulates the in vivo conditions. Because of that, the cells composing the sphere can be proliferative, non-proliferative, or necrotic based on their localization in the 3D structure. At present, spheroids have been developed using various cell types, such as hepatocytes [12], mesenchymal stem cells (MSCs) [13], neurons [14], lung cells [15], pre-osteoblasts [16], and several cancer cells [17,18,19].

Organoids are a more complex model and usually derive from the self-organization of induced pluripotent stem cells (iPSCs) or tissue-derived cells, such as stem and/or cancer cells [5]. Organoids are characterized by high complexity, given by their heterogeneous composition in terms of cell types, thus better reflecting the organ architecture and functionality [20]. Thanks to their complexity, they allow for long-term culture, which is not feasible with spheroids. Several organoids have been established in vitro to date, including pancreas organoids [21,22], thyroid organoids [23], gastric organoids [24], liver organoids [25], brain organoids [26], lung organoids [27], and retina organoids [28].

Thanks to the advances in new bioengineering techniques, a promising new model has been developed: the organ-on-a-chip. This model was created by combining biology and engineering knowledge to provide an advanced tool for the scientific community. The organ-on-a-chip is a microfluidic cell culture chip that mimics the activities, mechanisms, and physiological responses of entire organs, thus representing an in vitro artificial organ model [29]. Although this 3D model has been demonstrated potentially to substitute animal models in physiological and drug testing studies, the lack of standardization limits its applications. At present, the organs reproduced using this technique are the heart [30], lung [31], liver [32], kidney [33], bone [34], and skin [35]. Interestingly, Shuler’s research group has also developed the body-on-a-chip, using different microfluidic devices to integrate multi-organ activities and study multi-organ interactions [36].

For its unique features and growing application potential, 3D cell models are increasingly being proposed as valid alternatives to in vivo models in the study of physiological and pathological processes, as well as in pharmacological response [37]. The use of animals in experimentation must consider various aspects of complexity, including the high costs of management and care of laboratory animals, the ethical implications related to animal welfare in experiments, especially in relation to the possible causes of their suffering and fear, and the question of the transferability of results from animals to humans. In 1959, Russel and Burch wrote “The Principle of Humane Experimental Technique”, in which they formulated the 3R (Replacement, Reduction, and Refinement) principle [38]. Some years later, the European Directive n. 63/2010 (EU 63/2010) introduced the concept of non-animal-based methods, therefore increasing the need to develop valuable alternative models without the use of animals in research. In this context, 3D models attracted the interest of the scientific community since they are considered to be a bridge between in vitro and in vivo models and can reduce the number of animals used for experimental tests. In this review, we discuss, first, the most-used techniques for obtaining 3D cell cultures. We then describe some of the applications of these models, focusing on physiological processes, disease modeling, and regenerative medicine. The application of the 3D models to safety assessment, drug screening, and development is also discussed. Finally, we consider the challenges related to 3D cell cultures and future perspectives. 

## 2. Methods to Establish a 3D Cell Culture

The clear potential of 3D systems to provide new models suitable for studying cell interactions in both basic and more specialized research, revolutionizing cell culture technology, and offering alternative methods for animal experimentation, has prompted the scientific community to develop different efficient methods to establish 3D cell cultures, all of which, in turn, affect 3D model characteristics [39]. These techniques can be divided into two major categories: scaffold-free systems and scaffold-based systems. Scaffold-free systems are based on the self-aggregation capability of some cell types, which can be encouraged using specific cell plates and/or physical parameters that avoid cell attachment. On the other hand, in scaffold-based systems, cells are seeded in natural or synthetic materials, allowing cell proliferation, aggregation, and 3D organization.

### 2.1. Scaffold-Free Methods

#### 2.1.1. Pellet Cultures

One of the simplest methods of culturing cells in 3D is represented by pellet cultures, in which cells are pelleted to the bottom of the tube through centrifugal force (Figure 1). The supernatant is removed, while the pellet is suspended in a spheroid culture medium, and cells are seeded in multi-well plates with a cell-repellent surface [40]. It has been demonstrated that this technique can stimulate the chondrogenic differentiation of MSCs [41]. Using pellet cultures, dental pulp spheroids, bone marrow, and endothelial spheroids were also cultured [42,43].

#### 2.1.2. Liquid Overlay

The liquid overlay technique allows the growth of cells in 3D by using non-adherent surfaces, typically coated with hydrophilic and neutrally charged polymers (agar or agarose gel). This coating prevents the attachment of cells to the plate or flask, thus encouraging the cells to interact with each other and produce their own ECMs (Figure 2). The formation of aggregates can be forced by continuous agitation and/or centrifugation. With this method, it is possible to establish 3D cell cultures containing one or more cell types, with a spherical morphology and a variable size (50–150 μm) [44]. Human iPSCs were seeded in 96-well ultra-low attachment plates to create a model of multicellular human liver organoids [45]. The liquid overlay technique was successfully employed to obtain other 3D cell cultures, such as neural organoids [46], fibroblast spheroids [47], cancer spheroids [48], and MSC spheroids [49].

#### 2.1.3. Hanging Drop Method

Another scaffold-free technique to culture cells in 3D is the hanging drop method, in which cells are placed in a suspended drop of medium, thus allowing them to aggregate and form spheroids at the bottom of the droplet (Figure 3) [50]. Once the cell suspension is prepared, a drop of the culture medium containing the desired number of cells is dispensed into the wells of a mini-tray, and then the mini-tray is inverted upside-down, while the drop containing the cell suspension remains attached to the mini-tray by surface tension. This method takes advantage of surface tension and gravitational force to form a 3D cell aggregate in the droplets, and it is possible to control the size of the spheroid since it depends on the size of the drop and the concentration of the cell suspension [13]. Bartosh et al. used hanging drop methods to culture MSCs in 3D, demonstrating that MSCs grown in 3D secreted more anti-inflammatory molecules than those grown in 2D [51]. Other groups established hepatic spheroids [52], cancer spheroids [53], a model of tumor angiogenesis [54], mammary fibroblasts spheroids [55], and pancreatic islet spheroids [56] through this technique.

#### 2.1.4. Magnetic Levitation Method

In the magnetic levitation method, cells are mixed with a solution of magnetic nanoparticles and subjected to a magnetic force [57]. The magnetic nanoparticles are incubated with the cells overnight to allow their internalization. Then, the cells are detached and seeded in low-adhesive plates. A magnet is placed on the top of the plate lid and produces a magnetic force, which causes the cells to levitate against gravity, promoting cell–cell contacts and, therefore, leading to cell aggregation (Figure 4). By using this system, it is possible to culture different cell types, such as [58], MSCs [57], hepatocytes [59], cancer cells [60,61], and osteoblasts [62].

### 2.2. Scaffold-Based Methods

The development of regenerative medicine based on engineered biomaterials offers a plethora of scaffolds that can be used as a template for tissue formation [63]. These scaffolds must possess specific characteristics, such as chemical composition, shape, structure, and porosity scaffolds, in order to promote cell migration, adhesion, and tissue production [64]. Additionally, for a biomaterial to be considered applicable in tissue engineering, it must meet the following requirements, which include high biocompatibility, reactivity to cell adhesion, biodegradability, elasticity, and minimal toxicity. The shape, porosity, and surface morphology are also important aspects to create a suitable scaffold that can represent the real architecture of the tissue to be repaired and replicate its vascularization and multicellularity of the tissues [65], ensuring better transport of mass of the nutrients and oxygen [66]. Hydrogel scaffolds, such as gelatin and Matrigel, are natural [67,68] or synthetic [69] crosslinked hydrophobic biomaterials that can absorb and retain water, thus reproducing the natural ECM in compositional and structural terms [63]. Hydrogels have been used for various applications, such as for unicellular or multicellular 3D cultures to regenerate or study healthy or diseases tissues [70,71], or for the development of tumor spheroids [72,73] to study the tumor microenvironment. Commonly used scaffolds for 3D culture include naturally derived matrices and synthetic materials, which are discussed below.

#### 2.2.1. Natural Scaffolds

Natural scaffolds are represented mainly by decellularized scaffolds or composed of typical components of the ECM, such as collagen, elastin, laminin, or fibrin. Scaffolds made from natural compounds are biodegradable and biocompatible and can promote cellular interactions, adhesion, and signaling [74]. However, some of them showed major limitations, such as inconsistent purity resulting from batch-to-batch variability and difficulty in sterilization and purification. To find solutions to these limitations, tissue engineering continually searches for natural biomaterials from plant and animal sources, such as silk fibroin, chitosan, alginate, gelatin, or Matrigel, or from the glycosaminoglycan family, such as hyaluronic acid, heparin, derman sulfate, chondroitin sulfate, or heparan sulfate (Figure 5) [63].

#### 2.2.2. Synthetic Scaffolds

Synthetic scaffolds are represented mainly by polyethylene glycol (PEG), polyvinyl alcohol (PVA), polylactide-coglycolide (PLG), polycaprolactone (PCL), poly L-lactic acid (PLLA), poly (ethylene glycol) diacrylate (PEGDA), poly lactic-co-glycolic acid (PLGA), polytetrahydrofuran (PTHF), polyurethane (PU), and polyethylene terephthalate (PET); or by ceramics, such as calcium phosphate biomaterials, bioactive hydroxyapatite, and tricalcium phosphate or their associations (HA/TCP) and metals (Figure 6) [75,76]. All these materials meet the criteria needed for tissue engineering because they have higher biocompatibility, excellent biodegradability, and minimal toxicity.

### 2.3. Three-Dimensional Culture in Dynamic Conditions

#### 2.3.1. Bioreactors

One of the main limitations of using static cell culture conditions to obtain 3D models is the exchange of nutrients, as cell aggregates can reach 1–2 mm of thickness, making it difficult for the transfer of gases and waste products [77]. Bioreactors allow for the establishment of dynamic 3D cell cultures, by controlling several parameters of the extracellular microenvironment, such as pH, temperature, flow rate, oxygen, nutrients, and waste products [77]. Several designs for bioreactors are available, including rotating wall vessels, direct perfusion systems, hollow fibers, spinner flasks, and mechanical force systems. The spinner flask technique, for example, consists of seeding cells into spinner flask bioreactors, where the cell suspension is continuously mixed by stirring [78]. It is crucial to choose the right convectional force of the stirring for the spheroid formation—if the force is too slow the cells will lay on the bottom of the container and not aggregate; conversely, if the force is too high the spheroids will be damaged [40]. This system has been employed in dynamic cell cultures of MSCs, resulting in a high grade of adipogenesis and osteogenic differentiation [79]. Another type of bioreactor is the rotating wall vessel, in which cells are subjected to microgravity by constant circular rotation [80]. This constant rotation keeps the cells in suspension, thus modulating the differentiation capabilities of the MSCs. Sheyn et al. found that in microgravity conditions, adipogenic differentiation is favoured over osteogenic and chondrogenic differentiation [81]. Figure 7 shows a schematic example of a bioreactor used to obtain 3D cell cultures.

#### 2.3.2. Microfluidic Systems

Finally, one of the most sophisticated methods to culture cells in 3D is represented by microfluidic devices, also known as organ-on-a-chip. Microfluidic systems are composed of microwells connected by microfluidic channels, thus allowing the continuous infusion of nutrients and growth factors (Figure 8) [82]. This is a key characteristic of microfluidic devices, as it addresses the main limitation of static 3D cell cultures, which is the inhomogeneous concentration of oxygen and nutrients in the cell aggregate. The resulting dynamic microenvironment reflects the in vivo conditions faithfully. This method has allowed the culture of hepatic spheroids with higher viability than those obtained with static culture conditions [83]. Moreover, tumor spheroids were found to have higher resistance to drug treatment when cultured in flow conditions compared with static ones [84]. Using microfluidic systems, it is also possible to co-culture different cell types, as demonstrated by Sun et al., who set up a model for drug screening by culturing spheroids composed of tumor cells and fibroblasts [85].

## 3. Applications

As discussed above, 3D cultures achieve high cellular complexity and accurately mirror the in vivo conditions. Spheroids and organoids paved the way for research aimed at identifying valuable methods to study both physiological and pathological processes. Moreover, 3D models are emerging as promising tools to develop preclinical test systems as alternative to animal testing. Over the years, many excellent in vitro tissue and organ models have been developed, although further efforts are still needed. Although animal experimentation remains the gold standard for preclinical tests, growing evidence highlights that the 3D models can closely mimic disease patterns in vitro, thus representing a suitable system to study the pathophysiological features as well as to investigate new potential treatment options. In this section, we discuss the main applications of 3D systems to obtain in vitro organ and disease models and to develop therapeutic approaches, from drug screening to regenerative medicine, which have been also schematized in Figure 9.

### 3.1. Three-Dimensional Cell Cultures as Organ and Disease Models

#### 3.1.1. Bone

The macroscopic role of the musculoskeletal system is structural, providing the physical scaffolding for the human body [86]. In particular, bone can be considered as a composite material with a specialized organic–inorganic architecture [87], responsible also for maintaining mineral homeostasis and retaining the intrinsic capacity for regeneration in response to injury, such as during skeletal development or ongoing remodeling throughout adult life [88]. Healthy bone physiology includes a coordinated set of bone modeling and remodeling events [89], such as the regeneration process to replace bone tissue that has been damaged or lost due to trauma, injuries, cancer, or congenital defects [90,91]. 

Unfortunately, bone trauma is a very common injury and can affect anyone at any age, with repercussions on society due to the loss of productivity as well as costs borne by the health system for surgical interventions, possible re-hospitalization, and physical rehabilitation of patients [92]. In addition to bone, the degeneration of articular cartilage due to traumatic and degenerative injury or pathological conditions, such as osteoarthritis and rheumatoid arthritis [93], can lead to the progressive loss of articular cartilage and osteochondral interphase, representing an important disability that requires a multidisciplinary response from the scientific community [94]. For these reasons, in vitro 3D models have become important tools in the development and testing of potential treatments and strategies to improve the regeneration process of bone, cartilage, or osteochondral interphase. On the one hand, they are useful for studying and improving the integration of the newly formed tissue with the surrounding environment [95]; on the other hand, they are considered valid 3D models for bone disease studies [96]. Indeed, 3D composite bone tissue scaffolds with both mechanical stability and drug-delivery functionality are employed to study the drug-delivery properties of statins, biocompatibility, alkaline phosphatase activity, and osteoblasts activity in vitro for the treatment of osteoporosis [97]. Iordachescu and colleagues have realized a micron-scale bone organoid prototype (defined as a human trabecular organoid) to study the effects of microgravity, degenerative events, and related temporal events in bone remodeling, which cannot be reproduced using other technologies in vitro and in vivo [98]. Osteoarthritis studies, showed complex 3D models to reproduce the osteochondral interface, involving elements of the cartilage component and the well-vascularised cartilage-to-bone transition zone [99]. These engineered constructs are multiphasic structures with a particular rigid and porous section containing osteoblasts and endothelial cells, corresponding to the subchondral bone, a hydrated and viscoelastic section containing chondrocytes to reproduce the cartilaginous region [96]. The triphasic models are the ideal system for studying the complex cellular interactions in health or disease conditions affecting this area [100,101]. Caire et al., to study rheumatoid arthritis (RA), a chronic inflammatory bone disease, created 3D models with synoviocytes from RA patients, mixing them with Matrigel. They define it as an organoid, but this model is a spheroid because it does not fully reproduce the physiology of the involved tissue. Despite this, they demonstrated through 3D synovial spheroids how pro-inflammatory cytokines and mechano-transduction events enhanced the YAP/TAZ nuclear translocation and transcriptional activity, regulating the critical cellular responses involved in RA [102].

A model of choice to reproduce the in vitro bone tumor environment, such as osteosarcoma, is a 3D spheroid, which is able to mimic tumor micro-regions or micro-metastases [103] to study the response to chemotherapy [104,105,106] or gene therapy approaches [107]. Pierrevelcin et al. used 3D spheroids to evaluate, under hypoxic conditions, the cell–cell interaction between osteosarcoma and macrophage cells obtained from patients; to estimate the propensity of osteosarcoma cells to invade the ECM in which they are incorporated; and to test the effects of two drugs affecting potential proliferation and/or migration [108]. Lin and colleagues compared different 3D models of osteosarcoma, 3D hydrogels, and 3D spheroids, through a multiomics approach, showing a reduction in several pathways, such as the proliferation and anabolic pathways, and an increase in the catabolic pathway, with a strong increase in autophagy-related genes expression, suggesting an increased autophagy level [109].

#### 3.1.2. Brain

The study of the development of the nervous system (NS) and the brain, is one the greatest challenges for the scientific community, made difficult by the complexity of the human NS and by the limited accessibility of brain tissue of living organisms [110]. In view of this, 3D cultures provide a promising opportunity to explore further the human brain physiology. Since NS is such a complex structure, 2D systems fail to properly represent the key features of NS cells and their multiple interactions. In fact, it has been shown that a 2D support leads neurons to assume a different morphology compared with 3D cultures, showing a reduction in neurite extension, direction, and number [111]. Moreover, the complex synapses among neurons cannot be fully replicated in a 2D model, where neurons form single-plane interactions with near cells, due to the singular plane offered by the flask.

However, recent studies have shown that 3D neuronal cultures can reproduce human neuron features, such as transcriptional patterns [112] and neuron–neuron interactions, mirroring the in vivo synapses [113,114]. In addition to neuronal cultures, 3D models of other cellular components of nervous system tissue regulating brain homeostasis, such as glial cells, have been developed [115]. Several studies performed by using these models have shown how they also allow an in-depth investigation of the physiological astrocyte’s interactions with neurons [116,117] or a replication of the physiology of oligodendrocytes [118,119]. Moreover, Abud et al. set up a 3D brain organoid with iPSC-derived human microglia-like cells resembling the functional properties and transcriptome profile of human microglia [120]. Different from 2D cultures, 3D systems provide a platform where it is possible to recreate the complex microenvironment surrounding NS cells, including ECM and its interactions with neurons [121]. In addition, several studies have recreated vascularized brain organoids to study neurovascular interactions [122,123]. 

Recently, Cakir et al. set up an engineered human embryonic stem cell to recreate human brain organoids with a functional vascular-like system that resembles the vasculature in the early prenatal brain. In this study, the authors engineered human embryonic stem cells to express ectopically the human *ETS* variant 2, which contributed to forming a complex vascular-like structure, including several blood–brain barrier characteristics, increased expression of tight junctions, nutrient transporters, and trans-endothelial electrical resistance, thus representing a concrete model to study brain physiology in vitro [124]. One of the most interesting applications of 3D cultures concerns the study of the physiological processes orchestrating the neurological development and maturation. Interestingly, Sloan et al. generated brain-region-specific spheroids based on the neural induction of 3D aggregates of human iPSCs. By using this system, it was possible to highlight the role of specific molecular pathways (such as the one mediated by SMAD) and growth factors (such as EGF and FGF2) in determining different specific cell destinies, as well as to study the maturation of astrocytes, one of the later stages of human brain development [116,125]. In addition, the assembling of region-specific spheroids has been used to study cell migration and neural circuit formation in a structure called “assembloids” [14,126,127,128]. Other studies, focused on functional sensory input and motor output, have suggested that organoids can be used to study neural connectivity between the different regions of the brain [129,130]. In addition, organoids cultured at the air–liquid interface were able to show a great improvement in the survival and maturation of neurons, exhibiting the capability to develop long, dense bundles of axons with specific orientations [129]. 

Since 2D models have serious limitations and are inadequate platforms for the characterization of the physiological processes underlying the development and functionality of the NS, they cannot be considered appropriate for studying the complexity of nervous system disorders (NSDs). The simplicity of the 2D cultures limits their application as relevant models to study NSDs, often due to drastic alterations of the multiple interactions among NS cells and the complex surrounding microenvironment.

Indeed, it is known that changes in the composition of the brain ECM, as well as the altered interaction between neurons and glial cells (astrocytes, microglia, and oligodendrocyte), lead to an impaired brain function, eventually determining the evolution of psychiatric disorders, such as schizophrenia [131,132,133,134,135], depression [136,137,138], drug addiction [139,140,141], and even degenerative neurological diseases, such as multiple sclerosis (MS) [142,143,144,145,146], Alzheimer’s disease (AD) [147,148,149,150], and Parkinson’s disease (PD) [120]. Altogether, these considerations clarify why 3D systems represent a robust model to study the mechanism underlying NSDs [120,151,152]. Recently, it has been reported by Agboola et al. that brain organoids generated from human iPSCs were useful for the in vitro study of several neurological diseases, from viral infections, such as Zika virus, Herpes simplex virus, and Cytomegalovirus, to neurodevelopmental diseases, such as autism spectrum disorders, and even neurodegenerative diseases, such as AD and PD [153]. Although brain organoids already represent a novel and promising model for the study of NSDs, this system still lacks a fundamental component of the NS: vasculature. Since neurovascular interactions are key in several NSDs, such as AD [154], PD [155,156], and amyotrophic lateral sclerosis [157,158], developing neural organoids with ever-increasing multicellular complexity is the challenge for the near future.

#### 3.1.3. Heart

In comparison with the results obtained with 3D models of other organs, the development of heart spheroid/organoid models still lags behind. Although the use of human iPSCs allow researchers to produce easily high amounts of specific cardiac cell types, the currently obtained 3D models cannot be considered faithful in vitro models of the human heart, as they lack the structural and cellular complexity of the cardiac tissue [159]. Interestingly, a 3D microtissue system composed of cardiomyocytes, cardiac endothelial cells, and cardiac fibroblasts, the three major cell types of the heart, derived entirely from human iPSCs, can represent an excellent resource to study the differentiation of human heart cells in the development and consequences of heart disease or drugs in vitro. The choice of the appropriate ratios of iPSCs is essential to obtain functional cardiac tissue: the correct cell ratios comprise 70% cardiomyocytes, 15% endothelial cells, and 15% cardiac fibroblasts [160,161] or a 10:5.5 ratio when using cardiac progenitors and mesenchymal cells from human iPSCs [162]. In an interesting study, Drakhlis et al. described a method to reproduce the first steps of human cardiogenesis in vitro. They generated organoids thanks to the encapsulation of human iPSCs in Matrigel and combined this strategy with directed cardiac differentiation induced by WNT pathway modulation [163]. In recent years, the in vitro generation of 3D cardiac microtissues from human iPSCs has become an increasingly critical procedure for mimicking heart pathological conditions relevant to disease modeling [160,164]. Heart disease is the leading cause of death worldwide [165], thus representing a major global health concern, and the number of affected patients has doubled in the last 10 years [166]. Lewis-Israeli et al. demonstrated that human heart organoids platform can recreate complex metabolic disorders associated with congenital heart defects, as demonstrated by an in vitro model of pregestational-diabetes-induced congenital heart defects. In particular, the electrophysiological analysis showed the irregular frequency of action potentials in pregestational-diabetes-induced organoids, suggesting arrhythmic events. Secondly, metabolic assays for glycolysis and oxygen consumption revealed a decreased oxygen consumption rate and increased glycolysis. Moreover, a reduced number of mitochondria and a higher number of lipid droplets were revealed; this suggested a dysfunctional lipid metabolism and a more glycolytic profile. Finally, the immunofluorescence revealed a drastic difference and perturbation in the morphological organization of human heart organoids [159].

Systemic lupus erythematosus (SLE) is a chronic autoimmune disease with cardiovascular complications. Park et al. studied the effect of SLE serum on 3D cardiomyocytes spheroids derived from iPSCs. They treated spheroids with serum and anti-Ro autoantibodies. While serum alone did not affect the calcium signaling of spheroids, calcium signaling became unstable in response to the anti-Ro autoantibodies. Secondly, they observed that apoptosis markers, including the BAX/Bcl2 ratio, were upregulated in response to both treatments, but the anti-Ro autoantibody treatment showed higher levels of expression. The same trends were also observed for the fibrosis and hypertrophy markers. Furthermore, the expressions of both caspases 3 and 8 were increased in the spheroids treated with anti-Ro autoantibodies [167]. Another useful model to study heart disease is the organ-on-a-chip. Liu et al. [168] were able to replicate the cardiac hypoxic microenvironment of ischemia and follow the action potential changes over time in the cell culture using patterned electrodes in a heart-on-a-chip device. The authors generated a microfluidic channel able to induce a rapid modulation of medium oxygenation, which mimicked the temporary coronary occlusion and played a role in the activation of the hypoxia-inducible factor-1-alpha (HIF-1α) pathway, a transcriptional regulator which facilitates the metabolic adaptation to hypoxia. Furthermore, Kong et al. [169] were able to recreate the increased ECM stiffness using a photopolymerizable hydrogel, mimicking the pro-fibrotic conditions. The exposition to a biochemical stimulus, such as transforming growth factor beta (TGF-β), could even more closely mimic the fibrotic microenvironment of the heart, and the strain-dependent cardiac myofibroblast activations were found after 7 days of mechanical compression, which also strongly relied on the myofibroblast maturity. The authors discovered, indeed, that mechanical compression and TGF-β played synergetic roles in phenotypic remodeling [169]. Moreover, Wang et al. used cardiac fibroblast overpopulation of the tissues to mimic a fibrotic model, avoiding the pleiotropic effects of TGF-β [170]. These two methods can be combined into microfluidic devices in additional adjacent compartments where endothelial cells can be integrated, and more complex, integrative models can be made.

#### 3.1.4. Liver

The liver is a multicellular organ comprising large lobes divided into lobules. Hepatic functions include metabolizing drugs, filtering blood, and producing bile and blood plasma proteins, such as albumin. Consequently, the liver plays an essential role in several physiological processes which are vital for each of us. That is the reason why a robust in vitro liver model that resembles the in vivo microenvironment is highly warranted to overcome the limitations of the 2D cultures. At present, several studies use hepatic 3D cell cultures to investigate liver physiology, using a system more closely resembling and representative of the hepatic microenvironment compared with a 2D culture monolayer. In addition, the complexity of the liver can be well represented using organoids. Hepatocyte organoids, indeed, displayed solid aggregations accompanied by a gain of hepatic polarities and improved hepatic functions. Liver organoids provide a useful tool for studies on liver development and regeneration thanks to the inclusion of cells and the organization in a representative hepatic microenvironment. For example, embryonic liver development depends on the essential interplay of several signaling pathways [171]. Liver organoids obtained from the iPSCs represent a model for studying physiological liver development [172]. Using liver organoids, Asai et al. discovered that paracrine signals derived from mesenchymal or endothelial cells promoted the maturation of iPSCs-derived liver organoids, which were able to summarize well the interactions between stromal and epithelial cells during liver development. Furthermore, the hepatocyte-like cells exhibited microvilli on the surface opposite the monolayer, and each cell was connected to adjacent cells with a desmosome-like structure. This was confirmed by immunostaining of the tight junction protein ZO-1 too [172,173]. In the healthy adult liver, most hepatocytes proliferate minimally. Peng et al. showed that tumor necrosis factor-α (TNFα), a well-known inflammatory cytokine, promoted the expansion of mouse hepatocytes in a 3D cell culture and allowed a long-term culture for more than 6 months. Single-cell transcriptome analysis revealed the expression of hepatocyte-specific markers (albumin, apolipoprotein A1, transthyretin, hepatocyte nuclear factor 4 alpha, keratin 8/18, and tight junction protein 1), proliferation markers (ki67, Cyclin A2, and Cdk1), as well as liver regenerative factors and transcription factors (Rela, Stat3, Yap1, Jun, Fos, and Myc). Moreover, the enrichment of specific functional classes, such as cytolysis, lipoprotein particle remodeling, blood coagulation, retinol metabolic process, regulation of G1/S phase transition, and response to tumor necrosis factor, was observed. Furthermore, hepatocyte epithelial markers were highly expressed [174]. Tostoes et al. demonstrated that hepatic spheroids were viable, functional, and stable over at least 35 days [175] and could preserve patient-specific phenotypes and functions for several weeks. In this work, immunofluorescence microscopy of human hepatocyte spheroids confirmed the presence of the liver-specific markers, hepatocyte nuclear factor 4α, albumin, cytokeratin 18, and cytochrome P450 3A. Moreover, the results showed that these spheroids spontaneously assembled a functional bile canaliculi network.

The 3D hepatic spheroids are described as suitable in vitro model for studying liver steatosis and facilitate the translational discovery of novel drug targets. Non-alcoholic fatty liver disease (NAFLD) has caught the attention of the scientific community due to its rapid increase in prevalence worldwide. NAFLD is a chronic and progressive liver disease caused by excessive triglyceride accumulation. This pathological condition can be associated with significant complications, such as steatosis, liver cirrhosis, and hepatocellular carcinoma [176]. Bell et al. cultured primary human hepatocytes (PHH) in a 3D spheroid configuration to obtain a model of human steatosis and insulin resistance which could mimic the human liver function in vitro. The proteomic profile of these spheroids was found to resemble closely the intact liver tissue. Furthermore, the authors reported that several pathways, such as glycolysis, gluconeogenesis, and the γ-glutamyl cycle, were found misregulated in the 2D but not in the corresponding 3D spheroid cultures [177]. Not only spheroids but also organoids can be used for NAFLD studies. The results by Ramli et al. were obtained by using hepatic organoids generated from iPSCs and adult liver stem cells to mimic the systemic abundance of lipids [178]. After incubating organoids with free fatty acids, a time- and dose-dependent increase in the number of cells with lipid accumulation and triglyceride levels as well as the increasing levels of reactive oxygen species and lipid peroxidation were observed. The analysis of the global gene expression also showed a clear increase in the expression of the genes related to lipid and carbohydrate metabolism and a general dysregulation of other metabolic processes. As explained above, NAFLD can induce pathological complications, such as fibrosis. Several results paved the way for the development of a valid 3D culture cell model to study fibrosis. A human 3D co-culture model of fatty hepatocytes (hepatoma C3A cells) and hepatic stellate cells (LX-2) allowed researchers to obtain spheroids which showed morphological and molecular hallmarks of altered lipid metabolism and steatosis-induced fibrogenesis, mirroring the human disease [179]. These fatty spheroids were also used to test the antifibrotic properties of sorafenib on steatosis-induced fibrogenesis, highlighting the potential use of this 3D in vitro model of NAFLD to support therapeutic and drug testing applications. The liver’s functionality also can be affected by microorganisms, and spheroids derived from human hepatocytes are valid models to study hepatitis C virus (HCV) infection and replication.

Scientific evidence shows that HCV entry into hepatocytes is mediated by the interplay among four receptors: Occludin (OCLN), Claudin 1 (CLDN-1), CD81, and Scavenger receptor class B member 1. Interestingly, spheroids cultures have been studied to evaluate the hepatic infection capability of different viruses. In a recent study, spheroid cultures of a hepatocyte-derived cellular carcinoma cell line (Huh 7.5 cells) and primary human hepatocytes expressing the viral entry markers were obtained and infected with HCV pseudoparticles [180]. These results highlight the importance of using 3D models to recapitulate the process involved in viral infection, which cannot be studied easily in a 2D model. Similarly, Fu et al. developed a 3D hepatocyte model able to be infected by the hepatitis B virus and to secrete virions after the infection. The authors evaluated the expression of different host factors essential for HBV infection and replication, such as the transcription factors retinoid X receptor A and HNF4A and the viral receptor NTCP. Further results demonstrated that the expression level of NTCP was elevated by more than 19.4 ± 2.2-fold in the 3D hepatocyte model after 10 days of differentiation relative to the controls [181].

Since the beginning of 2020, SARS-CoV-2 has been catching the attention of the scientific community from several points of view. One of them is the effect of SARS-CoV-2 on the human liver [182]. PHH 3D models permissive to SARS-CoV-2 infection have been obtained by inducing in hepatocytes the increase in the expression of the SARS-CoV-2 receptor ACE2 after the exposure to pro-inflammatory cytokines, such as interferons [183]. Furthermore, Yang et al. demonstrated that the human iPSC platform, able to generate multiple different cells and organoids, is permissive to SARS-CoV-2 infections, thus representing a valuable model to study the pathological effects of a SARS-CoV-2 infection in the liver [184]. Adult primary human hepatocyte organoids were inoculated with SARS-CoV-2, and analysis by qRT-PCR demonstrated a SARS-CoV-2 infection, as evidenced by the high levels of viral sgRNA transcripts of the replicating viral RNA. This was confirmed at the protein level by immunostaining for SARS-CoV-2 spike protein expression. Furthermore, the transcript profiling revealed the upregulation of chemokine expression, consistent with the profiles of tissues obtained after autopsies of COVID-19 patients.

#### 3.1.5. Lung

The lung represents a complex organ in terms of both structure and function. It is characterized by more than 40 cell types [185], which support an intricate and unique architecture, consisting of complex branching airways that terminate in an arborized network that includes conducting airways, bronchi, bronchioles, gas-exchanging units, and alveoli. This singular structure is required for the main function of the lung, which is the exchange of oxygen and carbon dioxide between the circulatory systems and the external environment. The complexity of this system can be lost when transitioning to a 2D cell culture platform. Since 3D culture models are emerging as powerful tools for the development of branched tissue and organs, to date, several studies have developed 3D organoids for the study of lung physiology [186,187]. Leibel et al. recently established a 3D whole-lung organoid from iPSCs, useful for the study of branching morphogenesis and maturation of the lungs [188]. The authors were able to mimic the embryonic development process of the lung by introducing growth factors and small molecules aimed at generating endoderm, anterior foregut endoderm, and, ultimately, lung progenitor cells. Finally, these cells were prompted to develop in 3D lung organoids in response to external growth factors. Eventually, the whole-lung organoids thusly generated, after the exposure to dexamethasone, cyclic AMP, and isobutylxanthine, were driven to early lung developmental stages, such as branching morphogenesis and maturation. Moreover, Miller and colleagues recently described a protocol to differentiate hPSCs and replicate the numerous and complex stages of lung maturation, including endoderm induction, anterior–posterior and dorsal–ventral patterning, lung specification, lung budding, branching morphogenesis, and, eventually, maturation [189]. To do so, the authors first seeded hPSCs in monolayers and directed cells to the endoderm and then to the anterior foregut endoderm. Once generated, foregut spheroids were cultured in a 3D Matrigel droplet, where they were directed to become human lung organoids and bud tip progenitor organoids, thus generating human lung organoids, possessing cell types and structures resembling the lung’s complex structure. Recently, lung-on-a-chip models are arising as cell culture devices that can replicate lung physiological gas exchange, which is crucial in determining lung cell viability and surfactant production [31]. Miller et al. recently designed a breathable lung chamber, capable of mimicking breathing by simulating gas exchange, contraction, and expansion of the lung organoids using a reciprocating pump, thus providing a system fully replicating the lung physiology and function [190]. In addition, in view of the key role that the interaction between lung tissue and the circulatory system plays in fulfilling the function of gas exchange, it is fundamental to replicate this interaction in vitro. Gas exchange between the blood and tissues is guaranteed by a network of capillaries, which are fundamental for nutrient delivery and cell waste removal [191]. Therefore, it is easy to understand why 3D organoids need a proper vasculature to assure oxygen and nutrient delivery to better recapitulate normal organ growth and function. Currently, the combination of microfluidic devices and lung organoids allows recreating the vasculature and air–liquid interface, thus simulating gas exchange and the complex in vivo microenvironment surrounding the lungs [31]. To date, 3D lung models have been applied in mimicking some parts of the deep respiratory tract [192,193]; however, beyond a further investigation of the lower respiratory tract, a possible future direction could be the exploration of the upper respiratory tract, i.e., nasal cavity, pharynx, and larynx. The 3D lung platforms also represent a suitable model to study lung pathology, providing a comprehensive system to discover new mechanisms underlying lung diseases. In support of this, Thacker et al. recently discovered a key role of alveolar epithelial cells in controlling Mycobacterium tuberculosis growth during early infection [194].

Various studies have employed a 3D platform for the modeling of lung diseases and revealed the suitability of the lung-on-a-chip systems in replicating multiple diseases, such as asthma [195], pulmonary edema [196], and pulmonary thrombosis [192]. Recently, Sachs et al. set up a long-term expanding human airway organoid, offering an alternative in vitro model for both chronic respiratory diseases and respiratory infections [197]. In this study, the authors generated airway organoids—one derived from cystic fibrosis patients, allowing for the assessment of CFTR function, and another from lung cancer resections and metastasis biopsies, allowing for the study of tumor histopathology, gene mutations, and drug screening. Moreover, they assessed that Respiratory Syncytial Virus infection increased organoid cell motility and preferentially recruited neutrophils upon co-culturing. In view of these results, the authors concluded that human airway organoids represented a versatile model for the study of hereditary, malignant, and infectious lung diseases. Lately, lungs have become protagonists of the current global emergency coronavirus disease 2019 (COVID-19), being the main target of the SARS-CoV-2 infection, which can cause lung complications, such as pneumonia and, in the most severe cases, acute respiratory distress syndrome (ARDS) [198]. Huang et al. set up a reversed-engineered human alveolar lung-on-a-chip model, capable of reconstituting the functional human pulmonary alveoli in vitro, making it possible to investigate ARDS from the SARS-CoV-2 infection [193]. Jung et al. recently set up a 3D respiratory epithelial tissue construct with a perfusable microvasculature. This system, recapitulating the key features of small airways and alveoli, provided a relevant platform to perform high-throughput microfluidic screening, useful for the study of ARDS [199].

#### 3.1.6. Skin

The skin is the body’s largest organ. Its function is to protect the organism from dryness, microbial infection, ultraviolet rays, and chemical compounds [200]. Skin is composed of three layers, the epidermis, the dermis, and the hypodermis, each consisting of different cell types, including keratinocytes, melanocytes, Langerhans cells, fibroblasts, and stem cells [201]. Since 3D skin models better resemble the natural architecture and functions of the skin than 2D models [202,203], they are an excellent alternative to animal model use, especially for testing cosmetic ingredients, an activity banned in animals since 2013. The first 3D skin models were obtained by seeding keratinocytes on collagen gels with an air–liquid interface, as described by Prunieras and colleagues [204]; however, since the stratum corneum, which represents the physical barrier of the skin, was not properly recapitulated in these models, to improve the system, it was necessary to combine keratinocytes with de-epidermized dermis or collagen gels populated with fibroblasts to obtain a more realistic tissue architecture, with lamellar body extrusion, and the formation of the stratum corneum [202,205,206,207,208]. Although 3D skin models faithfully reproduce the architecture of the organ, enough to be called “human skin equivalent” (HSE) [209], they still present some limitations due to the lipid composition hampering their use for permeation studies [210,211,212]. Nevertheless, the HSE may represent a better model to study skin appendages, such as nails, hair, and glands, whose development and composition can be hardly studied by using the 2D in vitro cell cultures. For instance, Lee et al. demonstrated that skin organoids, obtained by seeding iPSCs, naturally produced de novo hair follicles mimicking normal embryonic hair folliculogenesis [213]. Similarly, Tan and colleagues successfully reproduced human hair follicles by using heterotypic spheroids, demonstrating that keratinocytes played a key role in hair follicle development and function maintaining the compartmentalization between the dermal papilla and the surrounding dermal fibroblasts residing in the dermis [214].

The introduction of 3D skin models has provided to the scientific community a valuable tool better to study the pathological conditions which can affect the skin, including acne, alopecia areata, atopic dermatitis, psoriasis, Raynaud’s phenomenon, rosacea, and vitiligo [215,216]. For instance, the use of the 3D skin models co-cultured with immune cells, such as Langerhans cells, dendritic cells, T cells, and macrophages [217,218,219,220,221], finds application in the study of skin inflammatory diseases. Gruber et al. and Yuki et al. reported that the addition of a cocktail of Th2 cytokines (IL-4, IL-13, and IL-31) and TNFα to HSE reproduces the atopic dermatitis phenotype in vitro, well reflecting the pathological setting of the disease characterized by the impaired function of tight junctions [222,223]. Moreover, Bernard and colleagues showed that in 3D cultures of keratinocytes treated with IL-22, TNFα, IL-4, and IL-13, the expression of S100A7 and IL-13RA2 was increased, while FLG expression was decreased [224]. In a similar way, they and Sa et al. have developed psoriasis models using IL-20 subfamily cytokines, such as IL-19, IL-20, IL-22, and IL-24, or a mixture of IL-17, IL-22, and TNFα [224,225], which induced hypogranulosis, parakeratosis, and altered the expression of the genes involved in psoriasis. Recently Rioux et al. established an elegant 3D human immunocompetent skin model of psoriasis, which included T cells [226]. This model of complete differentiated epidermis was obtained by seeding T cells and keratinocytes in dermal sheets obtained by culturing for 28 days dermal fibroblasts derived from both healthy donors and psoriasis patients onto one dermal sheet. Using this model, the authors demonstrated that T cells impaired the morphology of the 3D skin model and increased the keratinocyte proliferation and the expression of S100A7, elafin, and keratin 17, markers of the disease, thus mimicking the main features of psoriasis. Finally, they placed the dermal sheet with T cells under the dermal sheet with keratinocytes and cultured them for other 3 weeks at the air–liquid interface to obtain a complete differentiate epidermis [226]. Three-dimensional skin models have also been employed to study the effects of microgravity on human skin spheroids formed by epidermal keratinocytes and dermal fibroblasts. Choi and colleagues found that microgravity reduced the diameter of the spheroids, decreased the dermal thinning, and induced changes in the gene expression compared with the normal gravitational environment [227]. Zhuang et al. investigated the aging process using a 3D organotypic culture model [228], developed by culturing pericytes, fibroblasts, and keratinocytes with an air–liquid interface [229]. They found that in aged skin, the number of dermal pericytes was reduced compared with young skin; they also found an age difference in the skin’s regenerative ability, which could explain the poorer healing of skin wounds with age [228].

#### 3.1.7. Cancer

Although historically, the traditional 2D in vitro cell culture system has been widely used in cancer research as the main preclinical tumor model, this system fails in properly mimicking the complexity of cancer biology. Tumor formation and evolution are shaped by two fundamental drivers: genetic changes in tumoral cells and the dynamic interplay between cancer cells and the tumor microenvironment (TME) [230,231]. However, a 2D culture system lacks these pivotal aspects, leading tumoral cells to diverge from their real properties and complex phenotypes. Gillet et al. showed that the traditional 2D culture condition selects cells with rapid proliferation, growth, and better survival rate, causing the upregulation of many genes involved in cell cycle regulation and primary metabolic processes when compared with primary tumors [232], thus leading the tumoral cells to become progressively less similar to their in vivo equivalents. As a result of this discrepancy, it has been demonstrated that 2D cell culture impairs several tumoral cells functions, such as cell invasion, apoptosis, and cell proliferation [233,234,235,236]. Beyond genetics, the interaction of cancer cells and TME is a key event in defining tumor cell behaviour. TME is the theatre of a complex network of interactions of tumoral cells with both ECM [237] and TME stromal cells, including cancer-associated fibroblasts (CAFs) [238,239], MSCs [240,241], macrophages [242,243], and T cells [244].

In 3D cultures, ECM can be recreated, thus enabling the mechanical and chemical sensing between cells and their environment [245,246], crucial in modulating cancer cell proliferation, differentiation, and apoptosis [239,247,248,249,250,251,252,253,254]. Two-dimensional models, due to their lack of these pivotal interactions, show critical differences in the proliferation rate of cancer cells and other cells of the TME, thus the development of a system mirroring the in vivo cell ratio is currently a challenge. Eder et al. showed that in the 3D spheroid co-culture of CAFs and advanced prostate cancer cells, the ratio of cancer cells to CAFs faithfully mimicked the in vivo condition [255]. On this basis, 3D cell cultures find application in the study of the reciprocal interaction between cancer cells and TME [256,257,258]. Norberg et al. demonstrated the mutual influence of pancreatic tumor cells and stellate cells in a 3D co-culture spheroid model. As a result of this interaction, cancer cells increased their proliferation and shifted towards a more mesenchymal phenotype; moreover, pancreatic stellate cells were activated into a myofibroblast phenotype [259]. One peculiar feature of TME is represented by hypoxia, a condition of low oxygen and nutrient levels that occurs in the core of in vivo tumors that reach a volume greater than 1–2 mm^3^. Hypoxia represents the leading stimulus of many cancer-related events, such as the upregulation of glucose aerobic metabolism [260,261], the activation of angiogenic growth factors and receptors, such as VEGF and VEGFR [262,263], the activation of epithelial–mesenchymal transition pathways [264,265,266], and the recruitment of pro-tumor immune cells, such as myeloid cells [267], regulatory T cells [268], and tumor-associated macrophages [269]. Despite its unquestionable relevance in tumor progression, the hypoxic condition of TME cannot be represented in a 2D system, in which cells are placed in a monolayer. Meanwhile, several studies have highlighted how 3D culture platforms, such as spheroids [270,271], organoids [272], 3D scaffolds [273], and microfluidic devices [274], can recreate this particular feature of TME.

In light of what has been described above, 3D cell cultures represent a novel high-quality model for the study of tumor biology and evolution. Cancer cell migration, for instance, has been traditionally studied through scratch/wound healing assays [275,276]. However, this assay has substantial limitations: it is possible, when performing the scratch, to damage the cells in proximity, thus influencing the end results. Moreover, it has been shown that 2D cultured tumoral cell motility significantly differed from that of 3D cultured tumoral cells [277]. For example, Hakkinen et al. showed that fibroblasts migrated faster in 3D cultures as compared with their counterparts in 2D cultures [278].

Transwell assays partly overcome these limitations, enabling the study of cell migration in a 3D environment; however, the result is still influenced by the pore size of the membrane and by gravity. Today, the application of 3D microfluidic systems could provide a better model for the study of cancer cell migration and, consequently, tumor spreading [279]. Recently Goh et al. proposed a new model for the visualization and quantification of migrating cells in a 3D microfluidic plate filled with Matrigel, under different extracellular stimuli, such as nutrient gradient, cytokines, and co-culture with fibroblasts [280]. The 3D systems also represent a good model to study tumor invasion [281,282,283,284]; in fact, by its definition, “invasion” is the destructive and restructuring movement of cells through a 3D barrier. Härmä et al. described the dynamic reversion of polarized prostate cell spheroids into invasive cells, which, interestingly, revealed an upregulation of AKT and PI3-Kinase pathways, which are known to be pivotal for invasion [236]; most drugs targeting these pathways show efficacy against invasion processes but have a weaker efficacy in 2D culture models, again highlighting the closeness of the 3D models to in vivo systems [285]. Furthermore, the 3D system offers a high-quality model to study tumor vascularization, which involves a heterogeneous population of cells, including endothelial cells, immune cells, CAFs, and MCs [286,287,288]. Being of such a complex construction, 3D supports can represent a controlled tumor microenvironment of co-cultured cells, recreating the in vivo setting of tumoral vascularization [191,289]. Cui et al. set up a 3D microfluidic angiogenesis model, in which they recapitulated the immunosuppressive conditions and immune–vascular and cell–ECM interactions of glioblastoma. This model enabled the study of the role of M2-polarized macrophages in supporting angiogenesis in mouse glioma cell lines [290]. The use of this type of 3D device provides a tool for the development of a model mimicking the significant interaction between tumors and the circulatory system, which is fundamental in every stage of cancer evolution, from neovascularization to metastasis spreading. Figure 10 summarize the main tumor processes which can be properly represented and investigated through 3D cell cultures.

### 3.2. Toxicology Drug Screening

#### 3.2.1. Bone

Advances in additive manufacturing, bioprinting, and microfluidics have led to a surge in the development and manufacturing of microfluidic cell culture platforms and organ-on-a-chip devices, such as the bone chip system [291] and bone metastasis models [292], to create bone structure for pharmaceutical pre-clinical testing [293]. Mandatari et al. tested the effect of vitamin K2 on a human osteoblasts/osteoclasts 3D dynamic co-culture system that reproduced the bone microenvironment, observing how this vitamin improved the functions of osteoblasts isolated from osteoporotic patients. These 3D bone constructs represented a useful model to test in vitro a patient’s possible response to vitamin K2 [294].

Hui-Peng Ma et al. investigated the efficacy of celastrol (tripterine), a chemical compound isolated from the root extracts, on a microfluidic chip-based co-culture of fibroblast-like synoviocytes (FLS) indirectly co-cultured with osteoblasts and osteoclasts, realizing a rheumatoid arthritis model. Celastrol activity was confirmed by a decrease in the effect of bone erosion through the reduction in TRAP activity, the inhibition of abnormal proliferation and migration of FLS, the suppression of the HIF-1a signaling pathway/CXCR4 and the TLR4/NF-kB-mediated expression of matrix metalloproteinase-9, and the reduction in the activation of proinflammatory pathways [295]. Sakolish et al. used a Ewing Sarcoma (ES) bone-tumor tissue chip to screen anticancer drugs, such as doxorubicin, cisplatin, methotrexate, vincristine, dexamethasone, or a combination of cisplatin, methotrexate, and doxorubicin (MAP). This realistic tumor model allowed for the testing of drug–tissue/device interactions by better characterizing drug kinetics, demonstrating that cisplatin, vincristine, and MAP were most effective in killing ES cells [296].

#### 3.2.2. Brain

To date, despite the sustained research, the therapeutic treatment for NSDs is quite limited due to a small number of approved compounds and due to their minor effect, often offering short-term results. The development of drugs for the treatment of NSDs includes preclinical testing in animal disease models, which, unfortunately, often show poor translational outcomes when entering human clinical trials. The 3D brain systems may be proposed as alternative preclinical models to overcome this gap. The NSDs show a great complexity both in aetiology and progress due to the key role of several neuronal interactions, such as neuron–ECM [297,298], neuron–astrocytes [299], and neuron–microglia [300], each of which can be a target for developing novel therapies. These crucial relations can be fully represented in 3D brain organoids, which can provide, in this light, a relevant model for drug screening. In addition, the 3D platform offers an innovative system to study drug delivery by recreating the blood–brain barrier (BBB) [301] and investigating the capability of the NS drugs to cross it [302]. In a recent study, Boghdeh et al. applied a human BBB organ-on-a-chip to evaluate the effectiveness of small molecules against Venezuelan equine encephalitis virus. The study revealed the therapeutic effects of omaveloxolone in preserving the BBB integrity and decreasing the viral and inflammatory load, picturing this as a robust model to investigate the ability of molecules to cross the BBB and their effectiveness [303]. 

#### 3.2.3. Heart

As discussed above, 3D engineered microtissues can be used potentially for in vitro disease modeling, and biological mechanistic studies are proposed as suitable models for drug testing [304]. Since these models, due to the presence of tissue-like properties, such as multiple cell interaction and cell–extracellular matrix interactions, exhibit the physiological characteristics of the myocardium and blood vessels, they represent excellent protoypes to study the cardiotoxicity of several drugs. Ravenscroft and colleagues showed that the co-culture of hPSC-CMs with primary cardiac fibroblasts and endothelial cells was associated with a more adult-like response to pharmacological agents compared with single cell-type cardiomyocyte in vitro models. Two negative (atenolol, lapatinib) and two positive (digoxin, dobutamine) inotropic compounds were used to identify whether one specific cell type was responsible for the differences observed. The results obtained by Ravenscroft and colleagues showed that the co-culture of hPSC-CMs with primary cardiac fibroblasts and endothelial cells allowed for a better prediction of the inotropic effects of the drugs, demonstrating the contribution of cardiac non-myocyte cells in cardiotoxicity [305]. In an interesting study, Yeh et al. demonstrated that the treatment of prevascularized human cardiac organoids obtained by co-seeding human cardiomyocytes with cardiac fibroblasts and endothelial cells with Molidustat, a selective prolyl hydroxylase domain enzymes inhibitor [306], significantly improved the endothelial network formation that was at least partially attributed to HIF-α stabilization and the upregulation of VEGF secretion. In this study, it was also shown that Molidustat treatment improved the survival of cardiac organoids exposed to both in vitro hypoxic and ischemic conditions [307]. Finally, novel microfluidic organ-on-a-chip models have been introduced and have led to better recapitulation of crucial organ-level functions, multicellular microarchitecture, and environment dynamics, providing a technological platform contributing to the development of suitable high-throughput platforms for cardiovascular drug development [304].

#### 3.2.4. Liver

Hepatocyte spheroids found an interesting application in drug-induced liver injury (DILI) studies thanks to their proprieties of being more representative than the monolayer of in vitro 2D cultures. In an interesting study, Bell et al. demonstrated that PHH spheroids were a proper system to study chronic DILI. They evaluated five hepatotoxins for 4 weeks of drug treatment. PHH spheroids were dosed with amiodarone, bosentan, diclofenac, fialuridine, and tolcapone, and the viability was determined after 2, 8, and 28 days. For all five hepatotoxins, prolonged exposure led to increased toxicity. This illustrated the potential of the PHH spheroids to detect hepatotoxicity for compounds generally negative in several in vitro studies [177]. Furthermore, being able to express higher levels of cytochrome P450 and other phase II enzyme activities, the organoids showed a better response to apoptotic drugs and could correctly metabolize molecules, such as rifampicin, omeprazole, phenobarbital, and paracetamol [308,309], allowing for the discovery of several drug-adverse effects in the human liver [310,311]. Romualdo et al. studied the effect of sorafenib, a multi-kinase inhibitor that showed antifibrotic effects in the liver in animal models [312,313] and in steatosis and fibrosis induced in a human 3D co-culture model of NAFLD [314]. The treatment with sorafenib significantly reduced the viability of fatty spheroids at 48 and 72 h. Moreover, the transcriptional modulation of the genes related to lipid metabolism in fatty spheroids was different from that observed in their non-fatty spheroids. Sorafenib increased the mRNA levels of lipid oxidation- and hydrolysis-related genes. On the other hand, genes known to be upregulated in fibrosis, such as collagen type I alpha 1 chain, platelet-derived growth factor, and the tissue inhibitors of metalloproteinases, were downregulated significantly in sorafenib-treated fatty spheroids, demonstrating a clear sorafenib-mediated downregulation of fibrogenesis-related genes. In addition, sorafenib downregulated pro-inflammatory cytokines, such as IL-6, TGF-β1, and TNFα, in fatty spheroids [179]. The lack of a robust cell culture system permissive for hepatitis virus infection has limited its virus research and drug discovery. Recent evidence has showed that hepatosphere cultures could represent suitable in vitro models for studying the virus infection, for screening novel antiviral agents, and for testing anti-viral therapies [181,183,315]. Stebbilg et al. identified the antiviral and anti-cytokine efficacy of baricitinib, which is a Janus kinase-1/2 inhibitor, against SARS-CoV-2. In particular, the effects of the IFN-α2-mediated induction of ACE2 were evaluated on a viral load, and it was found that IFN-α2 increased the viral copy numbers in 3D liver organoids. Exposure to baricitinib fully abolished the ACE2 induction by IFN-α2 and efficiently blocked the increased infectivity in cytokine-treated 3D liver cultures. In addition, the genes strongly induced by the IFN treatment were significantly downregulated by baricitinib, such as chemokines (CCL8 and CXCL10), major histocompatibility complex components (CD74, LAG3, and LAMP3), and several IFN-induced protein family members (IFIT).

#### 3.2.5. Lung

Three-dimensional lung systems provide an accurate model recapitulating human physiology, employable for toxicology studies [31,316] and allowing, as far as possible, the reduction in the application to animals. Recently, Zhang et al. carried out nanotoxicity studies on a 3D human lung-on-a-chip model. This study investigated the potential hazards of inhaled nanoparticles on human health, using a lung-on-a-chip model consisting of a complex system of a co-culture of human vascular endothelial cells and human alveolar epithelial cells in a Matrigel membrane. This system, recapitulating the key features of the lung alveolar–capillary barrier, can be applied in lung toxicology studies concerning not only nanoparticles but also the environment, food, and drugs [316]. Beyond toxicology studies, 3D lung platforms represent a suitable preclinical model for drug screening, providing a comprehensive system in which to discover new mechanisms underlying lung diseases and new possible therapeutic targets. Jain et al. recently set up a primary human lung alveolus-on-a-chip as a model of intravascular thrombosis. This model allowed for the testing of potential antithrombotic therapeutics, thus revealing itself as a potential preclinical drug development tool [192]. Moreover, 3D lung models can be also applied for the study of inhaled drugs toxicity, delivery, and pharmacokinetics. Inhaled drugs are used largely for the treatment of several respiratory diseases, representing a non-invasive treatment that provides a higher local drug concentration and delivery in the bronchial and alveolar tissues, allowing, at the same time, the drug to reach the bloodstream [317]. Sivars et al. set up a 3D human airway in vitro model, providing a reliable predictive system for the detection of respiratory toxicities of inhaled drugs targeting respiratory diseases [318]. Finally, 3D lung models can be suitable tools for the identification and screening of drugs to fight the SARS-CoV-2 infection. Currently, drug screening for COVID-19 is performed mainly in 2D cultured cell lines, which fail to capture the dynamics of the SARS-CoV-2 infection, which not only can affect the lungs but also has been observed in several other organs, such as the intestines [319], heart [320], and liver [321]. Recently, Han et al. identified novel SARS-CoV-2 inhibitors by using both hPSC-derived lung organoids and hPSC-derived colonic organoids, thus considering the gastrointestinal manifestations that can be present with or without the respiratory symptoms [322]. This comprehensive view, offered by 3D models, provides a novel and relevant in vitro platform for high-throughput drug screening against SARS-CoV-2, helpful in identifying new potential drug candidates for COVID-19 patients.

#### 3.2.6. Skin

The development of HSE allowed for a reduction in animal experimentation in cosmetic and industrial applications [323]. Moreover, several commercial skin equivalents, such as Epiderm™ [324], EpiSkin^®^ [325], and SkinEthic™ RHE [326], have been established, giving the scientific community a useful tool for basic research and toxicological screenings [210,327,328]. For instance, in 2015, Abaci et al. established an HSE-on-a-chip model for testing the effect of the anti-cancer drug doxorubicin, showing the direct toxic effect of this drug on keratinocytes [329]. In the last decades, the use of nanotechnology, as well as the need for the assessment of the safety risks associated with exposure to nanomaterials, has emerged [330]. In a recently published study by Chen and colleagues, a 3D epidermis model, called EpiKutis, was employed to analyze the toxicity of silver nanoparticles (AgNPs) on the skin [331]. The EpiKutis model was obtained by seeding keratinocytes on a permeable membrane of transwell chambers and culturing them at the air–liquid interface for 2 weeks. This model successfully recapitulated the human epidermis since it was composed of multiple layers: a basal layer, a stratum spinous layer, a stratum granular layer, and a stratum corneum layer. It has been demonstrated that the AgNP treatment had lower toxic effects in the EpiKutis model than in keratinocytes cultured in 2D, which could be explained by the reduced penetration of AgNPs in the 3D model compared with that in the 2D cell culture. Moreover, while the AgNP treatment induced oxidative stress and pro-inflammatory cytokine expression in 2D keratinocytes, there was no increase in intracellular reactive oxygen species (ROS), malondialdehyde (MDA), superoxide dismutase (SOD), or pro-inflammatory cytokines in the EpiKutis model [331]. Considering these results, 3D cell cultures provide better estimates of in vivo effects, thus representing a clear advantage over 2D cultures in toxicological studies.

#### 3.2.7. Cancer

To date, the in vitro screening of anti-tumoral drugs has been performed mainly in 2D culture systems. However, it is noticeable that several drugs that show efficacy in 2D systems are less effective or show resistance when tested in vivo [332,333]. As a result, drugs that are effective in in vitro experiments often have no or weak efficacy in actual patients [334]. For this reason, at present, there is a growing interest in the definition of 3D cell cultures that can work as more faithful models for drug screening. Regarding cytotoxic agents, which represent the oldest drugs used for cancer treatment, many studies have compared cancer susceptibility with cytotoxic agents in 2D and 3D cultures [335,336,337,338]. In the in vivo treatment, tumor and stromal cells were exposed to different doses of drugs due to the 3D distribution of different cells in the primary mass, unlike in the in vitro 2D systems, where all present cell types were exposed to the same drug concentration. The 3D systems overcame these limitations, thanks to their multilayer structure, in which cells arranged themselves at different levels. In a recent study, Li et al. described how spheroids of several cancer cell lines exhibited higher drug resistance to cisplatin than cells cultured in monolayers, showing a response comparable to the in vivo tumor treatment [339]. This aspect was crucial not only for chemotherapies but also for all those therapies based on photodynamic cytotoxicity. Sokolova et al. compared the relevance of a photosensitizer both in ovarian adenocarcinoma spheroid and in a monolayer culture. The results of this study showed how the photosensitizer poorly penetrated the spheroids, leading to an inhomogeneous accumulation of the photosensitizers on the surface of the spheroid, mimicking the distribution of the photosensitizer in the in vivo tumor [340]. Currently, one of the major challenges in oncology research is to understand and overcome the chemoresistance leading cancer cells to evade the efficacy of anti-tumor drugs. In recent years, growing evidence highlighted the crucial role played by cancer stem cells (CSCs) in hindering the cancer susceptibility to therapy; in fact, several works have described how targeting CSCs could represent an optimal tool to sensitize tumoral cells to chemotherapy [341,342,343,344]. In light of this consideration, several studies focused their attention on setting up in vitro 3D culture models to properly represent CSCs [345,346]. Recently, Wang et al. synthesized a 3D porous scaffold, composed of chitosan and hyaluronic acid, as an in vitro model of glioblastoma. The authors observed how CSCs cultured on these scaffolds maintained the expression of the CSC-related genes and showed a higher resistance towards alkylating agents compared with the monolayer-cultured cells, thus suggesting that 3D scaffolds better mimic the in vivo biological and clinical features of CSCs [347].

### 3.3. Regenerative Medicine

#### 3.3.1. Bone

To develop the 3D bone constructs, MSCs currently represent the most suitable cell type compared with autologous osteoblasts, embryonic stem cells (ESCs), and iPSCs, which have shown several application problems related to their low proliferative potential, risk of teratoma development, immunological incompatibility, genetic manipulation concerns, or ethical concerns [348]. Furthermore, MSCs meet the following characteristics: availability in large quantities, greater or lesser capacity for osteogenic differentiation depending on the tissue of origin, painless isolation methods, and use in autologous or allogeneic transplantation, in accordance with the guidelines of the Good Manufacturing Practice (GMP) [349]. The gold standard materials employed in bone tissue engineering are designed to mimic the real bone architecture in terms of composition, interconnectivity, mechanical stability, and porosity, reproducing the physiological conditions as closely as possible [87]. By adopting mathematical modeling methods, it is possible to design scaffolds produced by additive manufacturing processes, such as 3D printing techniques that alternate the use of materials and cells [350] in static or dynamic conditions [351], to simulate the conditions in vivo more rigorously. The compatibility of MSCs with these materials and the ability to promote the bone regeneration process have been widely demonstrated in vitro and in vivo. Three-dimensional in vitro studies have reported the relevance of mineral compounds added to natural ones [87,352] or synthetic scaffolds [353,354,355] as inducers of osteogenic differentiation in human bone-marrow-derived MSCs in vitro [356]. On the other side, several in vivo studies have reported that the use of composite biomaterials, such as HA/alginate or HA/collagen, improved, by approximately fourfold, the regenerative potential of endogenous MSC cells by increasing bone formation [357]. Similarly, other studies have reported that 3D systems functionalized with growth factors [358,359], angiogenic factors [360], miRNA [361], exosomes [362], antibiotics [363], and other compounds are considered to be an added value in promoting bone regeneration [358]. Several challenges remain related to the application of cell therapy or the use of tissue-engineered or functionalized scaffolds to promote bone regeneration in vivo. The promising research data produced to date also bode well for solving problems related to the translational applications of 3D bone constructs, such as standardization of the manufacturing and analytical processes.

#### 3.3.2. Brain

Despite what has been thought, our NS is capable of regeneration, even in adults. This process, called neurogenesis, is ascribed mainly to nerve stem cells, which have been observed in the subventricular zone of the lateral ventricles and the subgranular zone of the dentate gyrus in the hippocampus [364]. In the light of this recently acquired knowledge, regenerative medicine of the NS is emerging as a new challenge for the scientific world. Current studies showed that stem-cell-based therapy can offer beneficial effects for several NSDs, such as stroke [365,366,367], epilepsy [368,369], PD [370,371,372], and other chronic neurodegenerative disorders. Organoids containing neural stem cells and neural progenitor cells are proposed as rich sources of autologous cells and tissue transplantation. Revah et al. recently reported that human stem-cell-derived cortical organoids transplanted into the somatosensory cortex of newborn athymic rats underwent cellular maturation and integration into sensory- and motivation-related circuits [373]. The efficacy of human embryonic-stem-cell-derived cerebral organoids as a treatment of mild traumatic brain injury (TBI) in a mouse model, in terms of repair of damages cortical regions, neurogenesis, and improved cognitive function, has been also described. After transplantation, neuronal death was reduced, neurogenesis in the ipsilateral subventricular zone and dentate gyrus of hippocampus was promoted, and angiogenesis increased. As a result, TBI-related neuronal dysfunction was fixed [374]. Although there are still limited data available in the literature about brain organoid applications in replacement therapies, being such a rich source of NSC and NPC, brain organoids could represent a promising strategy to develop new approaches of regenerative medicine for several NSDs.

#### 3.3.3. Heart

Biological therapies using cells and tissue are intended to repair and regenerate the diseased heart by improving tissue structure and function. In recent years, the in vitro generation of 3D cardiac microtissues from human iPSC has become an increasingly critical procedure for modeling the development of healthy human heart tissue and regenerative medicine strategies [375]. For example, Bargehr et al. tested the hypothesis that cardiac fibroblasts would be optimally suited for tissue engineering and heart repair strategies. The authors tested the capability of human ESC-derived epicardium to increase the function and structure of engineered heart tissue and to improve the efficacy of hESC-cardiomyocyte grafts in infarcted athymic rat hearts. Additionally, the co-transplantation of hESC-derived epicardial cells and cardiomyocytes doubled the graft cardiomyocyte proliferation rates in vivo and improved the systolic function [376]. A recent study confirmed the feasibility of using iPSC-CM spheroids in cell therapy and regenerative medicine. In particular, it was shown that when the spheroids of cardiomyocytes obtained from iPSC-CM were fused together within the fibrin matrix and administered as a patch over the infarcted region of mouse hearts, the spheroid patch was rapidly vascularized, the engraftment rate was remarkably high (>25%), and the treatment was associated with significant improvements in the infarct size and cardiac function [377]. 

#### 3.3.4. Liver

The lack of transplantable organs represents one of the most important problems in the field of transplantation, thus the development of regenerative medicine caught the attention of the scientific community. The liver has a high regenerative capability, and, consequently, it is a proper candidate for regenerative therapy [378]. Liver organoids have the capability to imitate hepatocyte regeneration during liver homeostasis, and they are gaining importance in the field of regenerative medicine, being considered as biological alternatives for organ replacement [379]. Hu et al. showed that hepatocyte organoids grown for several months retained key morphological, functional, and gene expression features of proliferating hepatocytes after partial hepatectomy, and after engraftment into mice, they recapitulated the proliferative damage response of hepatocytes. The organoids were transplanted as single cells into immunodeficient Fah−/− NOD Rag1−/− Il2rg−/− (FNRG) mice by splenic injection. For the first 30 days after the transplantation of human albumin in mice, the circulation remained stable and was clearly detectable in all mice that received the organoids. After day 30, the organoid graft started to proliferate more rapidly and expanded. Additionally, the repopulating grafts stained positive for ALB, MRP2, and CYP2E1, which indicated their functional maturity [380]. In another study, Takebe et al. created in vitro liver buds (LBs) by co-culturing hepatic endoderm cells from human iPSCs with HUVECs and MSCs to mimic liver development. These LBs, mechanically stable and physically manipulable, showed an evident maturation and vascularization in vivo after transplantation into nude mice with liver injury. Moreover, LBs showed hepatic cord-like structures and an improved survival of the mice. This study underlined that LB strategy is a significant method for studies on organogenesis in vitro [381]. In 2019, Wang et al. developed liver organoids from human embryonic stem cells, called hEHOs, exhibiting a remarkable repopulation capacity in the injured livers of mice following transplantation. The significant reduction in serum markers of hepatocyte injury, such as alanine aminotransferase and aspartate aminotransferase, demonstrated a hepatic maturation of hEHOs and recovery of liver functions. Moreover, hEHOs-derived hepatocytes repopulated 20% ± 5.6% of the liver parenchyma in the surviving mice, and cells were positive for cytokeratin 18, a marker of liver function [382].

#### 3.3.5. Lung

Although, to date, only a few studies have explored this possibility, lung organoids represent a potential tool for tissue regenerative approaches [383]. Since it has been discovered that new lung growth can occur in adults [384], the scientific community is focusing on studying lung repair and regeneration as an emerging and promising strategy for the treatment of patients with both acute and chronic lung diseases, which can cause the loss of functional lung tissue, such as that occurring in chronic obstructive pulmonary disease [385], pulmonary fibrosis [386], CF [387], and pulmonary arterial hypertension [388,389]. Currently, the ultimate treatment in patients with severe end-stage chronic disease is represented by lung transplantation; however, due to the shortage of transplant organs and the rejection of the transplanted ones, it becomes essential to find alternative solutions for the treatment of advanced lung diseases. Lung regeneration has already contributed to improving the life quality of patients with advanced pulmonary disease, suggesting that more focus is needed to explore further this strategy, including with the support of 3D system applications. To date, several region-specific stem/progenitor cells have been identified as responsible for epithelial regeneration in the lung, i.e., basal cells in proximal airways [390,391], neuroendocrine cells [392], and variant Club cells [393] in bronchioles, bronchoalveolar stem cells in the bronchoalveolar–duct junction [394], and alveolar type 2 epithelial cells in the alveolus [395,396,397]. Tan et al. recreated a multicellular airway organoid derived from human adult stem cells, capable of self-organization and maturation toward lung-tissue-like structures. The authors performed a successful ectopic engraftment of the organoid, marking the first step towards a future application in lung regenerative medicine [398]. Recently, Shulimzon et al. developed a new approach for the treatment of lung disease treatment, based on a catheter-injectable hydrogel-based scaffold, dedicated to both remodeling the pulmonary architecture and regenerating the lost respiratory tissue. This novel procedure represents a promising strategy for lung tissue engineering, which could find clinical applications in lung-regenerative medicine [399].

#### 3.3.6. Skin

The progress of skin tissue engineering allowed for the development of permanent coverage for large and deep wounds. Several attempts, based especially on the use of stem cells, were made to regenerate wounds into functional skin [400,401]; however, some of them failed. In this context, 3D skin models have found wide use [402,403], so much so that companies developed commercially available skin substitutes, such as Integra^®^ [404], Biobrane^®^ [405], Dermagraft [406], and TransCyte [407]. The main limitations in the application of these skin substitutes are the formation of scars or hematoma as well as the accumulation of exudate [407,408,409,410]. Other HSEs, such as Apligraf [411] or OrCel [412], cannot be used in patients allergic to bovine products, since they are based on bovine collagen [413], while StrataGraft [414] can be used in these patients because is prepared using a non-bovine source of collagen. To overcome all the limitations carried by non-autologous models, Boyce et al. developed an autologous-engineered skin substitute [415]. They cultured keratinocytes and fibroblasts derived from the patients’ biopsies on scaffolds made with collagen and glycosaminoglycan [416]. The main advantage of this model is the requirement of a very small area of skin biopsy that is then expanded to cover a larger area of the body. Several clinical trials have been conducted using this model [417,418,419], thus highlighting the importance of 3D cell culture development in skin regenerative medicine. As mentioned above, the skin also possesses many appendages. Among them, sweat glands are essential for human survival since they help to balance the body temperature response and function as excretory epidermal appendages [420]. Unfortunately, patients with large skin injuries experience the destruction of the overall skin architecture and a loss of the appendages, which can intensely reduce the quality of their lives. In 2021, Sun et al. successfully developed sweat glands organoids, starting from human epidermal keratinocytes cultured in a 3D Matrigel system, with the aim of generating personalized regenerative therapy in patients with large skin defects [421]. The obtained sweat gland organoids, having Ca^2+^ activity and expressing ductal, luminal, and myoepithelial markers when injected subcutaneously in mice with deep dermal injury and the complete destruction of sweat glands, enabled de novo sweat gland morphogenesis, thus representing a promising model for in vivo tissue replacement therapy [421].

## 4. Concluding Remarks: Challenges and Future Perspectives

Thanks to the advent of 3D cell cultures, in vitro studies are closer to animal models in many aspects, and they offer the possibility to study the complex interactions that were not possible with 2D cultures. At present, 3D cell cultures have several applications: drug discovery, pharmacological studies, understanding cell physiology and pathology, gene and protein expressions, cancer research, and regenerative medicine.

As 3D cell cultures are increasingly being used, more advanced methods will be applied in order to provide the throughput needed for large-scale testing as well as rapid and cost-effective screening. Furthermore, the current challenges in microscopy due to the large dimension size of 3D cell cultures compared with 2D cell cultures and the ability of different compounds (antibodies, vital dyes, drugs, as well as the oxygen itself) to penetrate 3D structures will be resolved.

Three-dimensional models have been employed by many research groups in recent years to increase the knowledge of many biological processes; however, the field of 3D cell cultures still needs standardization. Current 3D models differ in terms of complexity, size, morphology, and culturing protocols, which challenge assay protocols as well as the phenotype and output for analyses. Additionally, a proper correlation between phenotypic features of cells in 2D and 3D cultures is still missing, thus limiting the potential applications of these innovative models. More studies focused on a direct comparison between cells grown in 2D and 3D cultures, with a correlative response to stimuli or drugs, are needed. Furthermore, the use of 3D cell cultures may be limited by the costs of the materials to maintain cells in 3D, the expertise required to manage these models, the reproducibility, and the identification of the right assay for analyses [6,40,422]. The assays used for 3D cell cultures are less standardized and developed compared with the assays for 2D cultures in terms of imaging, analysis, quantification, and, especially, automation.

Nevertheless, as described above, several studies demonstrated that 3D cell cultures represent promising tools to study both the physiological and pathological processes, as well as the pharmacological response, and are, thus, considered valid alternatives to in vivo models [37]. These findings increased the interest of the scientific community in 3D cell cultures, making them the possible bridge between in vitro and in vivo models. 

This review summarized the updated knowledge of 3D cell cultures, from the most common methods used to set up a 3D culture to their interesting applications. This growing field opens up new possibilities for in vitro studies with the promising aim to “replace, reduce, and refine” the use of animals for experimental procedures.

## Figures and Tables

**Figure 1 ijms-24-12046-f001:**
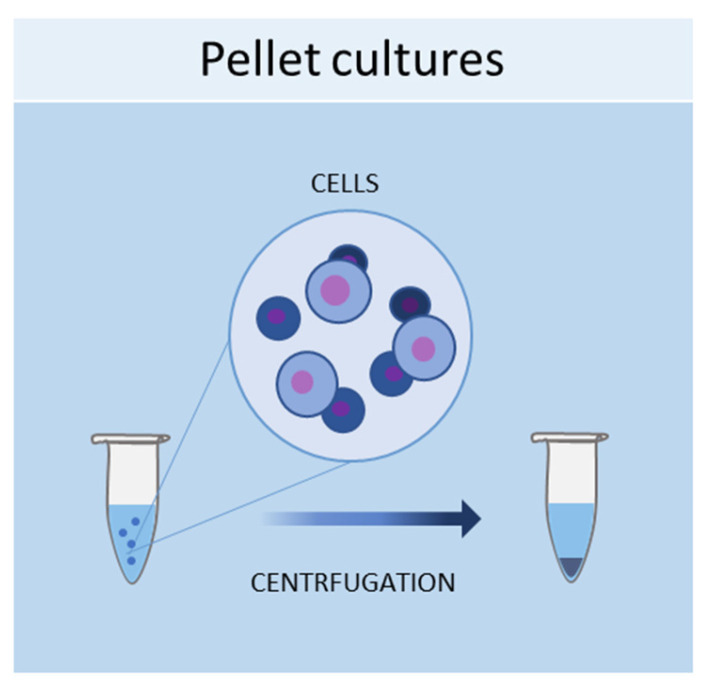
Schematic representation of pellet culture method used to obtain 3D cell cultures.

**Figure 2 ijms-24-12046-f002:**
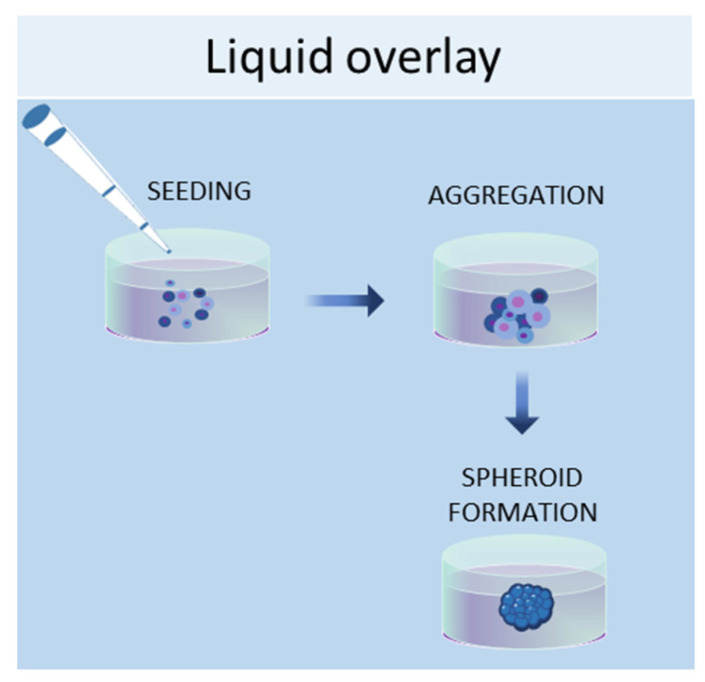
Schematic representation of liquid overlay technique used to obtain 3D cell cultures.

**Figure 3 ijms-24-12046-f003:**
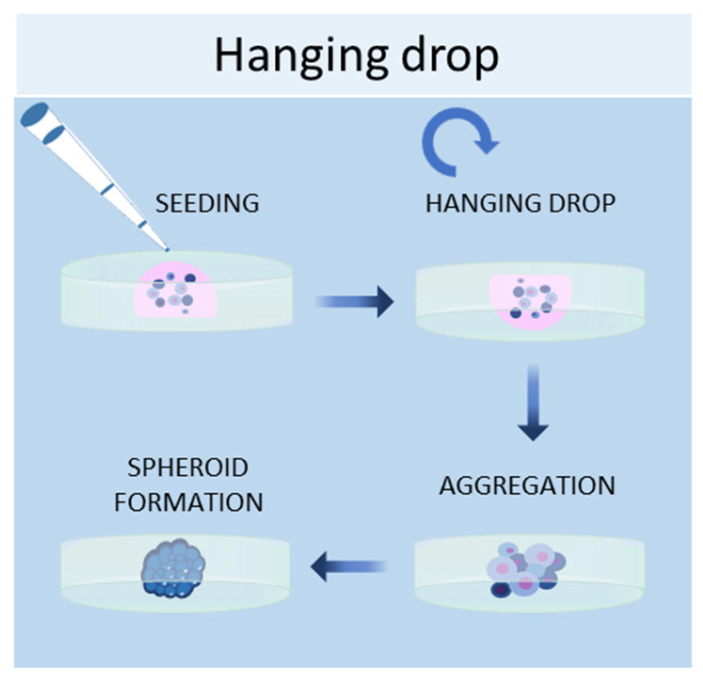
Schematic representation of hanging drop method used to obtain 3D cell cultures.

**Figure 4 ijms-24-12046-f004:**
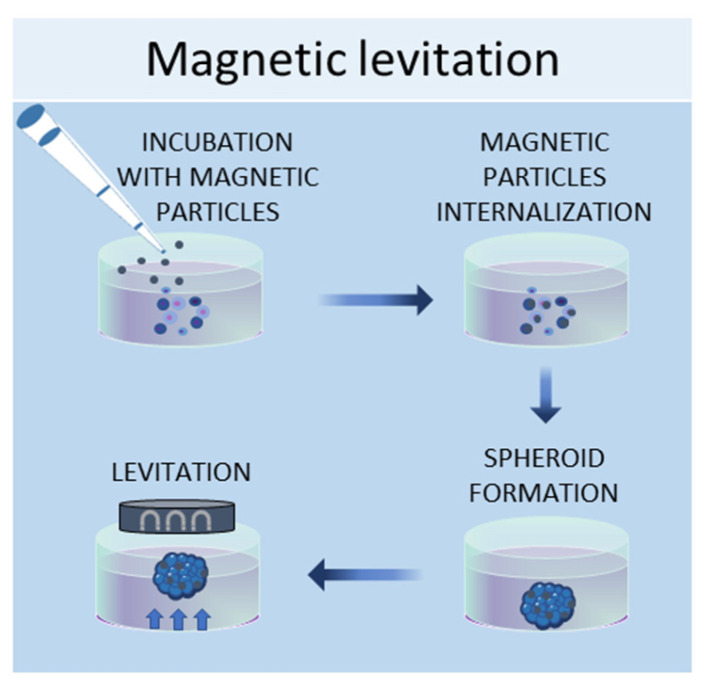
Schematic representation of magnetic levitation method used to obtain 3D cell cultures.

**Figure 5 ijms-24-12046-f005:**
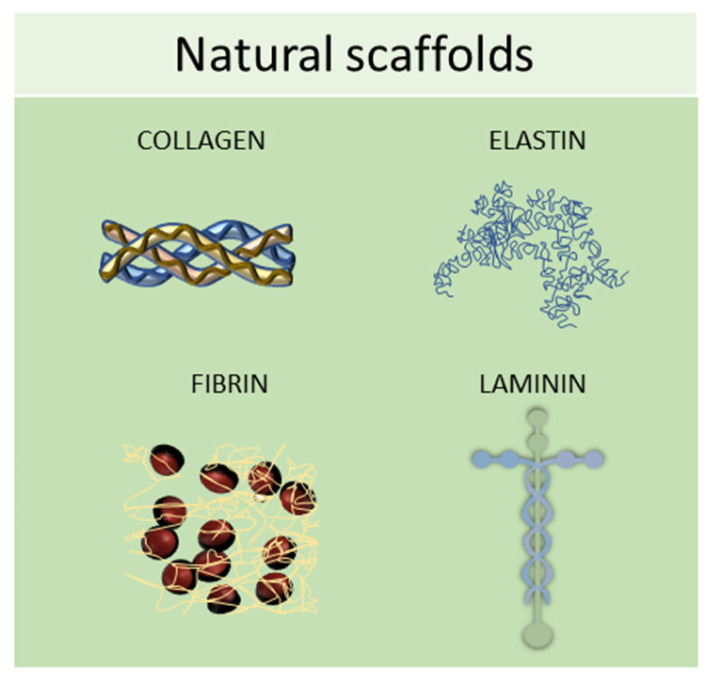
Schematic representation of some of the most used natural scaffolds (collagen, elastin, fibrin, and laminin) employed in scaffold-based 3D cell cultures.

**Figure 6 ijms-24-12046-f006:**
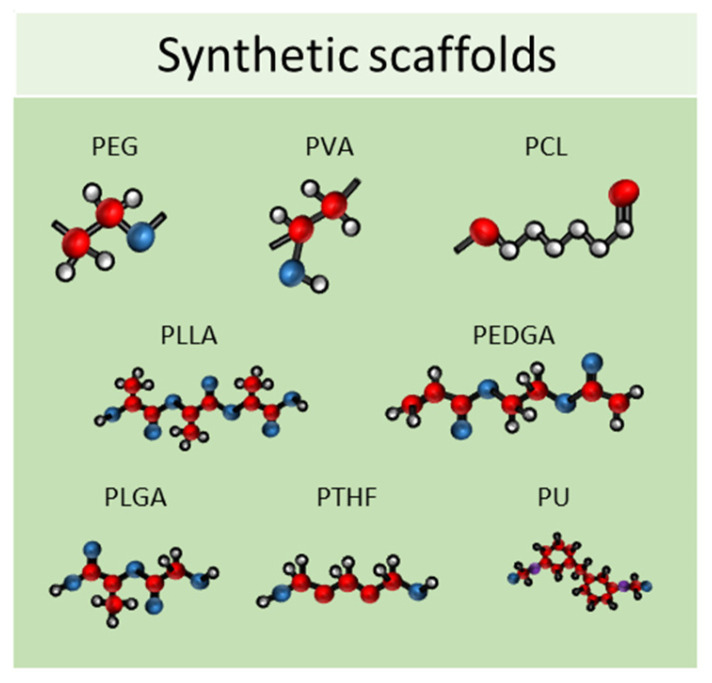
Schematic representation of some of the most used synthetic scaffolds used in scaffold-based 3D cell cultures. PEG: polyethylene glycol, PVA: polyvinyl alcohol, PCL: polycaprolactone, PLLA: poly L-lactic acid, PEGDA: poly (ethylene glycol) diacrylate, PLGA: poly lactic-co-glycolic acid, PTHF: polytetrahydrofuran, PU: polyure-thane.

**Figure 7 ijms-24-12046-f007:**
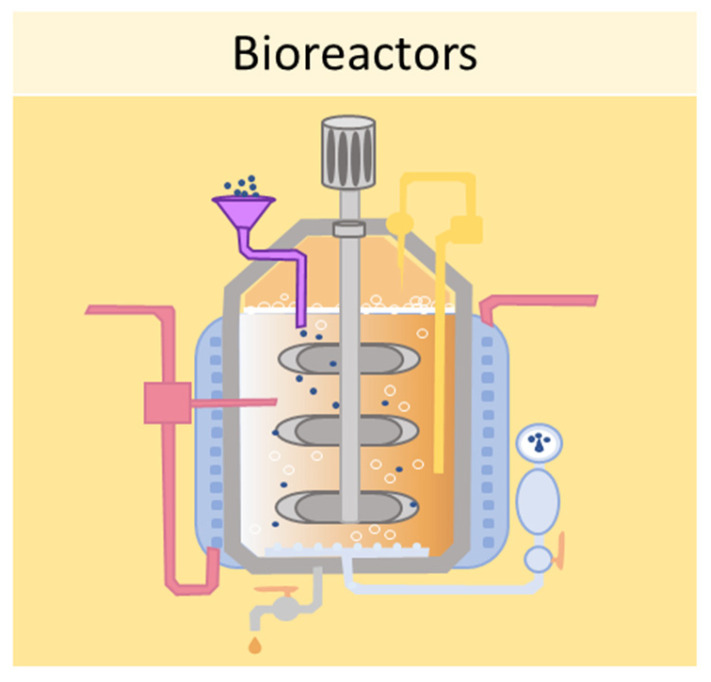
Schematic representation of a bioreactor which can be used to obtain 3D cell cultures.

**Figure 8 ijms-24-12046-f008:**
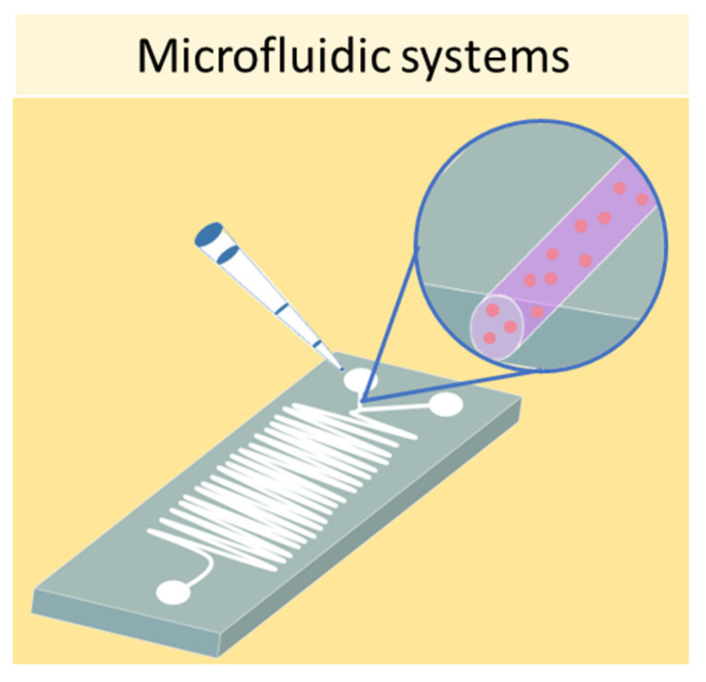
Schematic representation of a microfluidic device which can be used to obtain 3D cell cultures.

**Figure 9 ijms-24-12046-f009:**
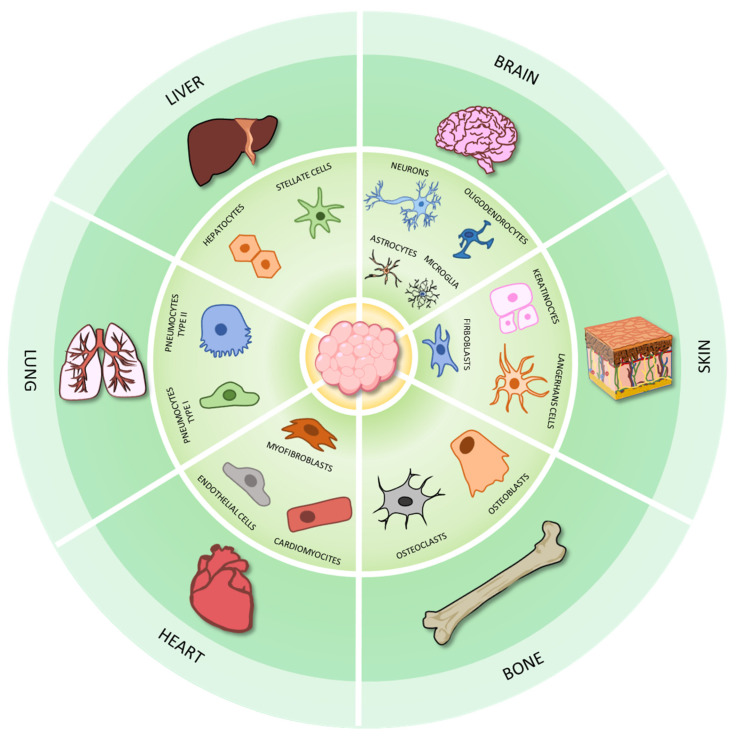
The applications of 3D cell cultures to model organ physiology. Three-dimensional models have been employed to model many organs and tissues, such as bone, brain, heart, liver, lung, and skin.

**Figure 10 ijms-24-12046-f010:**
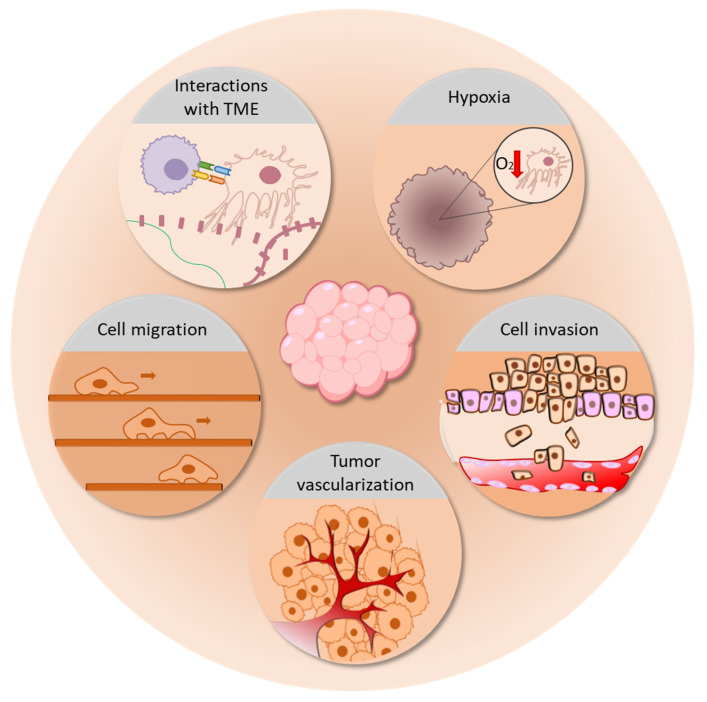
The applications of 3D cell cultures in tumor studies. The 3D models allow the study of many processes involved in tumor progression, including the interactions between cancer cells and TME, tumor hypoxia, cancer cell migration and invasion, and tumor vascularization.

**Table 1 ijms-24-12046-t001:** A comparison between 2D and 3D cell cultures.

2D Cell Cultures		3D Cell Cultures	
Advantages	Disadvantages	Advantages	Disadvantages
Inexpensive	Not representative of real cell environment	Better mimic tissue-like structures	Expensive
Well-established	Lack of predictivity	Exhibit differentiated cellular function	Reproducibility could be unsatisfactory
Large amount of comparative literature	Lack of cell–cell interactions	Simulate microenvironment conditions	Some systems are limited due to the static conditions
Easier cell observation and measurement	Lack of cell–ECM interactions	Better predict in vivo responses to drug treatment	Difficulties in finding the right assay for analysis

## Data Availability

No new data were created or analyzed in this study. Data sharing is not applicable to this article.

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
