# Peer review of "Three-Dimensional Cell Cultures: The Bridge between In Vitro and In Vivo Models"

_ijms, 2023, doi:10.3390/ijms241512046_

Round 1

Reviewer 1 Report

The authors of this manuscript offer a thorough review of the progress and potential of 3D cell culture systems in experimental research. They provide an overview of common 3D cell culture establishment methods and discuss their diverse applications, particularly in organ physiology, disease modeling, cancer research, drug screening, and regenerative medicine.

Overall, this review is comprehensive, well-written, and covers an important topic in biomedical research. The authors have done a commendable job of presenting the transition from 2D to 3D culture models and their benefits.

However, to fully convey the uniqueness and benefits of 3D culture systems, especially for readers who are less familiar with these methods, I recommend the authors to address the following points:

1.     In the comparison between 2D and 3D culture systems, it would be beneficial to address the technical challenges that may be associated with 3D cultures. For example, some experiments that are routinely done in 2D cultures, such as IF staining, may be more complex in organoids due to penetration issue.

2.     The authors could further emphasize the advantage that 3D cultures often allowing cell propagation without the need for immortalization. This can help maintain the integrity of critical tumor suppressor genes like ARF, INK4A, and TP53, which could be instrumental in research focused on tumor initiation.

3.     An additional point that could be included is the advantage of combining 3D culture with mouse models. For instance, if gene editing is carried out in ex vivo organoids, these can then be orthotopically transplanted into recipient mice. This method can lead to tissue-specific mutations without the need to generate a new transgenic mouse strain, which can save a huge amount of time.

4.     Some minor issues with the citation format. (line 109, line 291)

Author Response

Reviewer 1

The authors of this manuscript offer a thorough review of the progress and potential of 3D cell culture systems in experimental research. They provide an overview of common 3D cell culture establishment methods and discuss their diverse applications, particularly in organ physiology, disease modeling, cancer research, drug screening, and regenerative medicine.

Overall, this review is comprehensive, well-written, and covers an important topic in biomedical research. The authors have done a commendable job of presenting the transition from 2D to 3D culture models and their benefits.

We thank the reviewer for his/her positive remark on our work and the comments he/she gave us which have been very helpful to improve the quality of our manuscript.

However, to fully convey the uniqueness and benefits of 3D culture systems, especially for readers who are less familiar with these methods, I recommend the authors to address the following points:

  1. In the comparison between 2D and 3D culture systems, it would be beneficial to address the technical challenges that may be associated with 3D cultures. For example, some experiments that are routinely done in 2D cultures, such as IF staining, may be more complex in organoids due to penetration issue.

We thank the reviewer for his/her comment which allowed us to improve our manuscript. We agree with him/her and we added the sentence “Moreover, some exiting protocols for 2D cell cultures have to be revisited for 3D models, like immunofluorescence for instance in which problems related to the penetration of the staining as well as the clearance of the samples have been described” at lines 58-61.

  1. The authors could further emphasize the advantage that 3D cultures often allowing cell propagation without the need for immortalization. This can help maintain the integrity of critical tumor suppressor genes like ARF, INK4A,and TP53, which could be instrumental in research focused on tumor initiation.

We thank the reviewer for this point that allowed us to improve our work. We agree with him/her, and we added the sentence “Besides, 3D cell cultures allow cell propagation without the need for immortalization. This feature is essential to maintain the integrity of critical suppressor genes such as ARF, INK4A, and TP53, which are of interest in research focused on tumor initiation” at lines 48-52.

  1. An additional point that could be included is the advantage of combining 3D culture with mouse models. For instance, if gene editing is carried out in ex vivoorganoids, these can then be orthotopically transplanted into recipient mice. This method can lead to tissue-specific mutations without the need to generate a new transgenic mouse strain, which can save a huge amount of time.

We thank the reviewer for his/her comment which allowed us to improve our manuscript. We agree with him/her, and we added the sentence “Finally, another advantage of the use of 3D cell cultures is the possibility to combine them with mouse model, for instance performing the gene editing ex vivo in patient-derived organoids which then are injected into mice. This procedure will definitively reduce the amount of time needed to generate new transgenic mouse strains carrying tissue-specific mutations” at lines 52-56.

  1. Some minor issues with the citation format. (line 109, line 291)

We thank the reviewer for noticing these typos which have been fixed (lines 124 and 343 of the lasts version of the manuscript).

Reviewer 2 Report

The presented article provides an impressive review with a substantial number of cited literature references and is well-prepared. However, it requires some additional considerations:

11. I suggest changing the title from "3D cell cultures: the bridge between in vitro and in vivo models" to "3D cell cultures linking between in vitro and in vivo models" (a proposal for the authors to consider. I am not particularly fond of the word "bridge" in the title).

2 2. Lines 20 and 98 - In scientific papers, it is advisable to avoid phrases like "fascinating." Please remove them.

3 3.   Please check the journal guidelines regarding the formatting of literature references in the text. Shouldn't they be in the format: [X]?

4 4.  Please work on the text formatting, such as removing unnecessary spaces between sections and within paragraphs. Please justify the text.

5 5.  Some paragraphs begin with indentation, while others do not. Please format the text according to the journal guidelines.

6 6. The information provided in the subsections is very brief, and the authors do not precisely describe the findings of other researchers. They only provide general information such as "it has been demonstrated." Please expand the length of each subsection by referring to the researchers' surnames and describing the results of their conducted studies.

7 7.  I suggest splitting Figure 1 into smaller parts and placing each part under the corresponding subsection. This will facilitate the reader's understanding of the text and the diagram. Currently, the compilation of schemes is included in one figure without any references in the text. Please break down the figure into separate parts, place them in the appropriate locations, and refer to them in the text.

8 8.  Please proofread the text for editorial errors, e.g., line 291 has an extra period.

9 9.  There is no reference in the text to Figure 2 and 3. Please provide the necessary context, comment on the figures, and do not include them only at the end of the section. The figures are visually appealing but should be integrated into the text, connected to it, rather than "appended."

110.  Please expand the conclusions section as it is too brief considering the amount of information presented in the text.

111.   Please pay attention to the formatting of the references to ensure they align with the IJMS guidelines. It seems that the year should not be in parentheses. Please check the guidelines for authors for clarification.

I have no comments regarding the quality of the English language. It is written in a clear and stylistically correct manner. There are minor language errors in the text that should be corrected, but they are few and do not affect the quality of the manuscript.

Author Response

Reviewer 2

The presented article provides an impressive review with a substantial number of cited literature references and is well-prepared.

We thank the reviewer for his/her positive remark on our work and comments that helped us to improve the quality of our manuscript.

However, it requires some additional considerations:

  1. I suggest changing the title from "3D cell cultures: the bridge between in vitro and in vivo models" to "3D cell cultures linking between in vitro and in vivo models" (a proposal for the authors to consider. I am not particularly fond of the word "bridge" in the title).

We thank the reviewer for this point, however, as he/she surely knows the title of a manuscript is something that should grab attention and make people want to read further. We are fond of this title, which is not so different by the one proposed by the reviewer. For this reason, since the reviewer just propose to change the title of the manuscript, we prefer to keep the title as it was.

2 . Lines 20 and 98 - In scientific papers, it is advisable to avoid phrases like "fascinating." Please remove them 

We thank the reviewer for his/her comment. According to this comment we removed the word “fashinating” in the above-mentioned lines of the manuscript at lines 20 and 111 of the last version of the manuscript.

3 .   Please check the journal guidelines regarding the formatting of literature references in the text. Shouldn't they be in the format: [X]?

We thank the reviewer for noticing the discrepancy between our format of references and journal guidelines. We have updated the format of literature references according to the journal instructions.

4 .  Please work on the text formatting, such as removing unnecessary spaces between sections and within paragraphs. Please justify the text.

We thank the reviewer for his/her comment. We have now adjusted the text formatting according to his/her suggestions.

5 .  Some paragraphs begin with indentation, while others do not. Please format the text according to the journal guidelines.

We thank the reviewer for his/her comment. We have adjusted the text according to the journal guidelines.

6 . The information provided in the subsections is very brief, and the authors do not precisely describe the findings of other researchers. They only provide general information such as "it has been demonstrated." Please expand the length of each subsection by referring to the researchers' surnames and describing the results of their conducted studies.

We thank the reviewer for this point which allowed us to better clarify the findings of cited works in each subsection. Considering the amount of information and the already big length of the manuscript we specified, where necessary, the surnames of the authors that performed the experiments and we provided a better described of the results.

7 .  I suggest splitting Figure 1 into smaller parts and placing each part under the corresponding subsection. This will facilitate the reader's understanding of the text and the diagram. Currently, the compilation of schemes is included in one figure without any references in the text. Please break down the figure into separate parts, place them in the appropriate locations, and refer to them in the text.

We thank the reviewer for his/her comment. As suggested, we have splitted Figure 1 into Figures 1, 2, 3, 4, 5, 6, 7, and 8. Each figure has been placed at the end of the corresponding subparagraph and cited in the text.

8 .  Please proofread the text for editorial errors, e.g., line 291 has an extra period.

We thank the reviewer for noticing this typo which has been fixed.

9 .  There is no reference in the text to Figure 2 and 3. Please provide the necessary context, comment on the figures, and do not include them only at the end of the section. The figures are visually appealing but should be integrated into the text, connected to it, rather than "appended."

We thank the reviewer for his/her comment. We have now included in the text the references to Figures 2 (now Figure 9) and 3 (now Figure 10) (respectively line 311 and line 811-812).

  1. Please expand the conclusions section as it is too brief considering the amount of information presented in the text.

We thank the reviewer for his/her comment which allowed us to improve the quality of the conclusion section of our manuscript. We expanded this section according to his/her suggestion (lines 1189-1201).

  1. Please pay attention to the formatting of the references to ensure they align with the IJMS guidelines. It seems that the year should not be in parentheses. Please check the guidelines for authors for clarification.

We thank the reviewer for noticing again the discrepancy between our format of references and journal guidelines. We have updated the format of literature references according to the journal guidelines.
